# Selectivity Drives Productivity: Efficient Dataset Pruning for Enhanced Transfer Learning

**Yihua Zhang**[1,*]   **Yimeng Zhang**[1,*]   **Aochuan Chen**[1,*]   **Jinghan Jia**[1]   **Jiancheng Liu**[1]

**Gaowen Liu**[2]   **Mingyi Hong**[3]   **Shiyu Chang**[4]   **Sijia Liu**[1]

[1] Michigan State University, [2] Cisco Research,
[3] University of Minnesota, Twin City, [4] UC Santa Barbara
[*] Equal Contribution

## Abstract

Massive data is often considered essential for deep learning applications, but it also incurs significant computational and infrastructural costs. Therefore, dataset pruning (DP) has emerged as an effective way to improve data efficiency by identifying and removing redundant training samples without sacrificing performance. In this work, we aim to address the problem of DP for transfer learning, *i.e.*, *how to prune a source dataset for improved pretraining efficiency and lossless finetuning accuracy on downstream target tasks*. To our best knowledge, the problem of DP for transfer learning remains open, as previous studies have primarily addressed DP and transfer learning as separate problems. By contrast, we establish a unified viewpoint to integrate DP with transfer learning and find that existing DP methods are not suitable for the transfer learning paradigm. We then propose two new DP methods, label mapping and feature mapping, for supervised and self-supervised pretraining settings respectively, by revisiting the DP problem through the lens of source-target domain mapping. Furthermore, we demonstrate the effectiveness of our approach on numerous transfer learning tasks. We show that source data classes can be pruned by up to $40\% \sim 80\%$ without sacrificing downstream performance, resulting in a significant $2 \sim 5\times$ speed-up during the pretraining stage. Besides, our proposal exhibits broad applicability and can improve other computationally intensive transfer learning techniques, such as adversarial pretraining. Codes are available at https://github.com/OPTML-Group/DP4TL.

## 1 Introduction

The abundance of the training data has long been regarded as the key driver of the contemporary machine learning (ML) algorithms [1–3]. However, the untrimmed, ever-growing training dataset could not only introduce training biases that compromise the model performance [4–6], but also poses an almost insurmountable obstacle to high training efficiency [1, 2]. Therefore, understanding the impact of the training data and selecting the most critical samples has emerged as an important goal, collectively referred to as the problem of *dataset pruning* (**DP**).

Although the feasibility and the promise of DP have been unveiled in numerous applications, such as noisy data cleansing [7–9], continue learning [10–13] and active learning [14], greater emphasis is placed on the *in-domain* training setting, *i.e.*, the pruned training set share the similar distribution as the evaluation set. Examples of methods to condense the training dataset with lossless generalization (on the in-distribution testing dataset) include data influence functions [10, 15–23], model training dynamics [11–14, 24–28], and coreset selection [29–36]. In contrast, our work investigates the

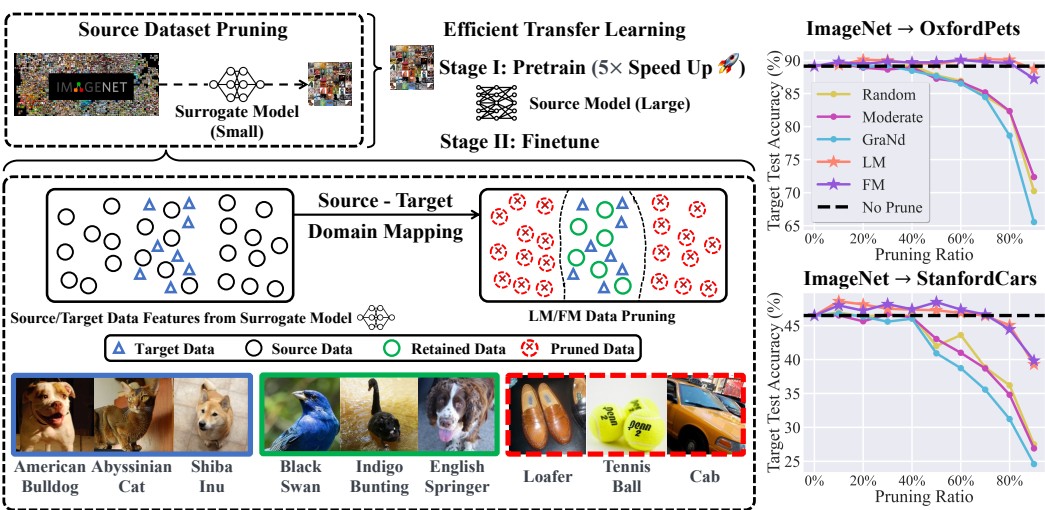

Figure 1: **Left**: An illustration of the proposed dataset pruning methods (LM and FM) and their performance overview. Large scale source dataset is pruned by LM and FM through a small surrogate model (ResNet-18). Large foundation models can achieve up to $5\times$ speed-up on pretraining without no downstream performance drop. **Right**: Downstream performance overview at different pruning ratios when ResNet-101 [54] is used as the source model (trained on the pruned source dataset) with ImageNet [55] as the source dataset transferring to OxfordPets [56] and StanfordCars [57]. The feature extractor is fixed during finetuning.

problem of DP in the **transfer learning** paradigm [37–39], which has emerged as a popular way to leverage the knowledge of a *foundation model* learned on a *source* dataset (referring to 'pretraining' stage) to further enhance the performance on a cross-domain *target* dataset (referring to 'finetuning' stage). Recent evidence [40–42] has shown that some source data points could make a *harmful influence* in the downstream performance. In particular, the previous study [41] showed that removing specific source data classes can improve the transfer learning accuracy of a pretrained model.

Yet, *efficiently* identifying those *harmful* source data classes for improved transfer learning is a highly non-trivial task. Firstly, the unique pretrain-finetune paradigm complicates the analysis of the influence of source data on downstream performance, making it an indirect and very challenging process. Consequently, existing gradient and data influence-based in-domain DP methods [10, 19–23] cannot be naïvely adapted to the transfer setting. Secondly, transfer learning encompasses a wide range of source training methods other than supervised learning, such as self-supervised learning (**SSL**) [43–52]. Therefore, a generic DP framework is desired given various pretraining scenarios. Thirdly, from an efficiency standpoint, the design of efficient DP algorithms is non-trivial. Even in the non-transfer learning paradigm, a majority of current in-domain DP methods [11, 25, 26] introduce substantial computational overheads. For instance, influence function-based DP methods [10, 15–22] necessitate the calculation of the inverse Hessian matrix of model parameters, which is a highly demanding and computationally intensive process. Moreover, training dynamics-based methods [11–14, 24–28, 53], such as GraNd-Score [26] and Forgetting-Score [11], necessitate model training on the entire dataset multiple times. As a result, the development of an efficient and effective DP framework tailored for transfer learning remains a significant challenge in the field.

The most relevant work to ours is [41], which proposed a brute-force method to evaluate the influence of every source class in the downstream task following a leave-one-out principle. While this method effectively selects the most influential source classes, it is prohibitively slow, as the data selection process demands significantly higher computational resources than performing transfer learning itself. This brings us to the central question addressed in this paper:

> *(Q) How can we extend DP to transfer learning with minimal computation overhead, broad applicability, and improved target performance?*

To address **(Q)**, we formally define the task of DP (dataset pruning) for transfer learning. We start by uncovering the limitations of conventional in-domain DP methods when applied to transfer learning tasks, and establish an intrinsic connection between source dataset pruning for transfer learning and source-target domain mapping. In the supervised pretraining paradigm, we propose an effective and scalable label mapping (**LM**)-based framework capable of pinpointing the most beneficial source

data labels for a cross-domain target dataset. Furthermore, we extend the idea of LM to feature mapping (**FM**) for the SSL pretraining protocol, where data labels are unavailable. To achieve a greater practical impact, we demonstrate that our proposed methods (LM and FM) can facilitate effective DP over a small, simple surrogate source model, and further translate the positive impact of the pruned source dataset to other larger, more complex source models. We provide a schematic overview and result highlights in **Fig. 1**. Our contributions can be summarized as follows:

❶ We connect the concept of DP (dataset pruning) to transfer learning for the first time and formalize the problem of DP for transfer learning.

❷ We develop a highly efficient and effective framework for DP in transfer learning, leveraging the source-target domain mapping with customized LM (label mapping) and FM (feature mapping) for both supervised and SSL (self-supervised learning) settings. Our approach is principled and achieves significant speedups compared to the state-of-the-art methods.

❸ We empirically show the effectiveness of our proposals (LP and FP) on 8 downstream tasks. We find that the source dataset (ImageNet) can be pruned up to $40\% \sim 80\%$ without sacrificing the downstream performance, together with a $2\times \sim 5\times$ speed-up in the pretraining stage. Our proposal also unlocks a door to prune a dataset using a simple surrogate source model (*e.g.*, ResNet-18) and then reuse the pruned dataset to improve transfer learning on a larger source model (*e.g.*, ResNet-101).

❹ Lastly, we show that our proposed DP framework can benefit other computationally-intensive transfer learning techniques. For example, DP for adversarial pretraining [58] leads to a $1\% \sim 2\%$ improvement in downstream performance with time consumption similar to standard pretraining.

## 2 Related Work

**Dataset pruning.**   DP (dataset pruning) is an emerging technique to improve the data efficiency of model training by selecting the most representative training samples or removing the less influential ones [10–28]. Thus, the problem of coreset selection [29–36] can also be considered a form of DP. Prior studies on DP range from clustering-based methods [59–61] to the more recent score-based pruning methods [10–28]. In the latter approach, an importance score is assigned to each training data point to quantify its influence on a particular permanence metric during model learning. Specifically, score-based DP methods can be broadly categorized into two main groups: influence function-based approaches [10, 15–23] and training dynamics-based approaches [11–14, 24–28, 53]. The first category measures data influence by examining the effect of data removal on the learning algorithm used during training [15–18] and the model's prediction [10, 19–23]. However, influence function-based approaches typically require high computational costs due to the need for high-order derivatives and complex optimization methods, such as bi-level optimization, as used in [22]. Although an approximate influence score can be obtained efficiently, it may result in a large estimation error [62]. In the second category, training dynamics-based approaches find statistical indicators for pruned data from the training trajectory. Examples of DP metrics include data loss/error [25–27], prediction confidence [24], model gradient norm [26], forgetting event [11], and compactness [12–14, 28]. However, these methods typically require repeated training to ensure the representativeness of the collected statistics [11, 25, 26]. In addition to improving data efficiency, DP has been applied in a variety of contexts, such as noisy label cleansing [7–9], continue learning [10–13, 63], active learning [14], and reducing annotation cost of a finetuner [64].

**Data valuation & attribution.**   Data valuation [19, 41, 65–70] and attribution [71–73] are research streams related to dataset pruning that aim to quantify the influence of training data points on model's performance. Unlike dataset pruning, these approaches do not focus primarily on training efficiency but instead aim to enhance interpretability [74], adversarial robustness [68, 70], generative model design [75], and data acquisition quality [72]. Representative methods include Shapley value-based methods [65, 66], datamodels [73], sampling-based methods [65, 68], and proxy-based methods [67, 72]. Recently, Kim et al. [35] proposed to select data samples from a large public dataset (Open-Set) for self-supervised learning given a specific target dataset through distribution mismatch. However, it fails to make a general framework in both supervised and self-supervised scenarios, leaving the former under-explored. The most relevant work to ours is [41], which leverages a leave-one-out analysis to quantify the influence of source data on downstream tasks in the context of transfer learning. With this inspiration, we propose to connect DP to transfer learning in this work.

**Recent advancements in transfer learning.** Transfer learning has been a prominent area over the past decade [37–39]. Significant strides have been made in understanding, analyzing, and improving various aspects of this technique [58, 76–84]. Recent studies [77, 78] have rigorously analyzed the relationship between source model performance and downstream task effectiveness, arguing that a narrow focus on minimizing source training loss may not lead to improved transfer learning results. To improve transfer learning, adversarial training has been shown to benefit the transferability of source pretraining on target downstream tasks [58, 76]. There also exist studies to identify and understand the failure cases in transfer learning [79–84]. Other research [77, 78] has examined model transferability from the perspective of model sharpness and argues that a good pretrained model should be situated in a flat basin in the downstream loss landscape. Last but not the least, transfer learning has progressed in several other directions, such as self-supervised learning [43–52], model weight pruning [85–89], and visual prompting [90–99].

## 3 Problem Formulation

In this section, we introduce some essential preliminaries on transfer learning and DP (dataset pruning), and elucidate the design challenge of DP for transfer learning.

**Preliminaries on transfer learning and connection to DP.** Let $g \circ f$ denote a deep neural network that consists of a feature extractor $f$ and a classification head $g$, where $\circ$ denotes the function composition. Given source and target datasets $\mathcal{D}_\mathcal{S}$ and $\mathcal{D}_\mathcal{T}$, we study transfer learning in the "*pretrain-finetune*" paradigm. The primary goal of *pretraining* is to obtain a high-quality feature extractor $f_\mathcal{S} : \mathcal{X} \to \mathcal{Z}$, which draws a mapping from the input space ($\mathcal{X}$) to the deep representation space ($\mathcal{Z}$) in a data-rich source domain ($\mathcal{D}_\mathcal{S}$). Popular pertaining recipes include supervised learning (**SL**) [84, 100] and self-supervised learning (**SSL**) [43–46] depending on whether the source labels ($y_\mathcal{S}$) are available or not in $\mathcal{D}_\mathcal{S}$. In the *finetuning* stage, the pretrained model is further trained on a specific downstream task under the target dataset $\mathcal{D}_\mathcal{T}$. Transfer learning expects improved downstream performance over training on $\mathcal{D}_\mathcal{T}$ from scratch. In this work, we consider two finetuning protocols, linear probe (**LP**) and full-finetune (**FF**), with $\mathcal{D}_\mathcal{T}$ being a labeled dataset. LP finetunes the linear classification head $g$ with a fixed feature extractor $f_\mathcal{S}$, acquired from pretraining. In contrast, FF finetunes the entire model $g \circ f$ from the initialization $f_\mathcal{S}$. FF typically yields a better transfer learning accuracy than LP, but the former takes a higher computation cost.

Our motivation for connecting transfer learning with DP comes from a recent data-based perspective on transfer learning [41]. The study shows that removing certain *source data classes* from $\mathcal{D}_\mathcal{S}$ can potentially improve the accuracy of a finetuned model on $\mathcal{D}_\mathcal{T}$. However, the task of evaluating the transfer effects of source data and removing their influence from a pre-trained source model has not been addressed efficiently. The approach developed in [41] involves a leave-one-out analysis to estimate the influence of a source class $c$ on a target example $t$, which is computed as the prediction discrepancy of the finetuned source model at $t$ when the class $c$ is either included or excluded from $\mathcal{D}_\mathcal{S}$. During this process, one must train multiple source models (over 7000 models on ImageNet in [41]) from scratch over different subsets of $\mathcal{D}_\mathcal{S}$ for a given target task. This approach becomes computationally unaffordable when dealing with large source datasets like ImageNet given limited computing resources. To address this challenge, we propose a DP perspective on transfer learning.

**Problem of interest: DP for transfer learning.** Next, we introduce the background of DP and the problem we focus on. Let $\mathcal{D} = \{\mathbf{z}_1, \mathbf{z}_2, \ldots, \mathbf{z}_N\}$ denote a dataset consisting of $n$ samples, where each sample $z_i$ is represented as a pair $(\mathbf{x}_i, y_i)$, with $\mathbf{x}_i$ denoting the input feature vector and $y_i$ denoting the corresponding label. DP aims to generate a pruned dataset $\hat{\mathcal{D}} = \{\hat{\mathbf{z}}_1, \hat{\mathbf{z}}_2, \ldots, \hat{\mathbf{z}}_M\} \subset \mathcal{D}$ with $M < N$, which can reduce the training cost without a significant decrease in model generalization performance when trained on $\hat{\mathcal{D}}$. In the context of [41], instead of individual source data sample $\{\hat{\mathbf{z}}_i\}$, the entire source classes are evaluated and selected in terms of transfer influences. Based on the above, we define the problem of our interest below.

> **(DP for transfer learning)** How to *prune* source data classes to obtain $\hat{\mathcal{D}}_\mathcal{S}$ (a subset of $\mathcal{D}_\mathcal{S}$), with lossless or improved transfer learning accuracy of the source model ($f_\mathcal{S}$) on a target task $\mathcal{D}_\mathcal{T}$?

DP for transfer learning has two key distinctions (❶-❷) from the vanilla DP setup. First (❶), DP must be performed in the source domain ($\mathcal{D}_\mathcal{S}$), while its effectiveness is evaluated based on the target

domain ($\mathcal{D}_\mathcal{T}$). This 'cross-domain' challenge makes the design of efficient and effective DP highly non-trivial. For example, prior work [41] utilizes a computationally-intensive leave-one-out analysis. Classical influence function-based methods [10, 19–23, 101], which trace the eventual model's prediction through the learning algorithm and back to its training data, are also computationally infeasible due to the complex bi-level optimizer and the calculation of high-order derivatives. Second (❷), the pre-trained source model ($f_\mathcal{S}$) in today's transfer learning regime is typically of a large scale. This motivates us to develop a DP method that can be independent of the source model, while the pruned source dataset $\hat{\mathcal{D}}_\mathcal{S}$ remains effective in the original transfer learning setup. Given this challenge, we will design DP methods for transfer learning using a *simple surrogate source model* to avoid the computation on the large source model $f_\mathcal{S}$ (see **Fig. 1** for an illustration).

**Conventional DP methods lack effectiveness on transfer learning.** Given the challenges (❶-❷) posed by DP for transfer learning, we further conduct a preliminary study to investigate the effectiveness of existing 8 DP methods, including SP [20], SSP [20], GRAND [26], EL2N [26], MODERATE [3], FORGET [53], INF-MAX [53], and INFSUM [53]. Our results show that these methods are *unable* to yield significant improvements over *random* source data pruning. **Fig. 2** shows the transfer learning performance of the ResNet-101 [54] model trained on different

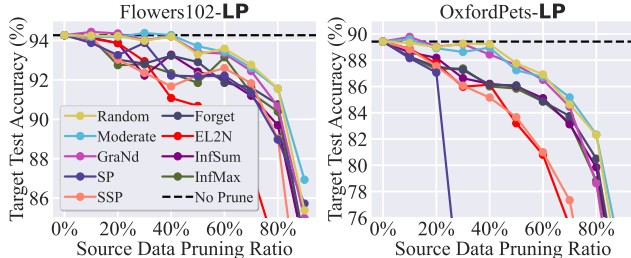

Figure 2: Transfer learning accuracy of existing DP methods on ImageNet at different pruning ratios, where ResNet-101 is the source model, and linear probing (LP) is used for downstream finetuning on the target datasets Flowers102 (**Left**) and OxfordPets (**Right**).

pruned versions of the ImageNet dataset when LP-based finetuning is conducted on the downstream Flowers102 [102] and OxfordPets [56] datasets. As we can see, in transfer learning, random pruning is a solid baseline for various state-of-the-art DP methods, which have demonstrated superior performance to the former in the non-transfer learning regime. Therefore, it is crucial to develop an efficient and effective DP method specifically tailored for transfer learning.

## 4 Label/Feature Mapping-based DP for Transfer Learning

In this section, we first introduce a simple yet powerful DP method called label mapping (**LM**) by leveraging the class-discriminative capability of a supervised source model. We then extend the concept of LM to feature mapping (**FM**), suitable for self-supervised pretraining.

**LM-based DP for supervised pretraining.** Following the notations used in Sec. 3, we represent the model obtained through supervised learning on the source dataset ($\mathcal{D}_\mathcal{S}$) as $g_\mathcal{S} \circ f_\mathcal{S}$. This model predicts a source label given an input example ($\mathbf{x}$). In DP for transfer learning, the source data classes serve as the variables to be pruned. Thus, we express $\mathcal{D}_\mathcal{S}$ as $\mathcal{D}_\mathcal{S} = \{\mathcal{C}_\mathcal{S}^{(1)}, \ldots, \mathcal{C}_\mathcal{S}^{(N)}\}$ for $N$ source classes, $\mathcal{C}_\mathcal{S}^{(i)}$ denotes the set of data points belonging to the source class $i$. Additionally, the pruner has access to the target dataset $\mathcal{D}_\mathcal{T} = \{\mathbf{t}_1, \ldots, \mathbf{t}_n\}$, which consists of $n$ target data points. Our objective is to utilize the information provided by $g_\mathcal{S} \circ f_\mathcal{S}$, $\mathcal{D}_\mathcal{S}$, and $\mathcal{D}_\mathcal{T}$ to devise a computationally efficient DP criterion. Importantly, our criterion should not involve model training, distinguishing it from the non-scalable approach in [41]. An important observation by [78] in transfer learning is that transferred knowledge improves transferability. This insight is supported by the loss landscape analysis in transfer learning: The finetuned weights may remain within the flat basin of the pretrained weights for enhanced transfer learning performance.

We next propose to extract the 'transferred knowledge' by leveraging the class-discriminative capability of the source model $g_\mathcal{S} \circ f_\mathcal{S}$ on the target data samples in $\mathcal{D}_\mathcal{T}$. Specifically, we focus on monitoring the responsiveness of the source label predictions made by $g_\mathcal{S} \circ f_\mathcal{S}$ when using target samples $\{\mathbf{t}_i \in \mathcal{D}_\mathcal{T}\}_{i=1}^n$ as input data. Here we resize these target samples to ensure their resolution alignment with source data. For the $i$th source data class $\mathcal{C}_\mathcal{S}^{(i)}$, we then define its pruning score below:

$$\mathbf{s}_{\mathrm{LM}}(\mathcal{C}_\mathcal{S}^{(i)}) = \sum_{j=1}^{n} \mathbb{1}(g_\mathcal{S} \circ f_\mathcal{S}(\mathbf{t}_j) = i), \text{ for } i = 1, 2, \ldots, N, \quad \text{(LM)}$$

where $\mathbb{1}(\cdot)$ represents an indicator function that evaluates to 1 when the condition $\cdot$ is satisfied, and 0 otherwise. We refer to the aforementioned formula as LM (Label Mapping) since the condition $g_\mathcal{S} \circ f_\mathcal{S}(\mathbf{t}_j) = i$ establishes a mapping between the predicted labels of the target samples $\{\mathbf{t}_j\}_{j=1}^n$ by the source model and the corresponding source labels $i$. The larger $\mathbf{s}_{\mathrm{LM}}$ in (LM) signifies that a specific source class is more frequently utilized to interpret the target data. Consequently, this indicates a more tightly connected relationship between the source class and the transferred knowledge. Therefore, we prune (or select) source data classes with low (or high) $\mathbf{s}_{\mathrm{LM}}$ values.

Although LM is simple, it offers several advantages. *Firstly*, it is highly efficient in computation. Given a pre-trained source model $g_\mathcal{S} \circ f_\mathcal{S}$, the calculation of LM only requires forward passes through the model. In cases where obtaining the pretrained model in the source domain is challenging, our proposal supports an alternative approach: training a *smaller and simpler surrogate model* to carry out LM. This surrogate model can effectively replace the complex pretrained model $g_\mathcal{S} \circ f_\mathcal{S}$ and facilitate the efficient execution of the pruning process. As we show in **Fig. 3a**, employing ResNet-18 [54] is sufficient to successfully prune the source dataset (ImageNet [55]). The resulting DP scheme remains effective to improve transfer learning utilizing other larger source models, such as ResNet-101 [54]. *In addition*, **Fig. 3b** shows that the computational overhead incurred

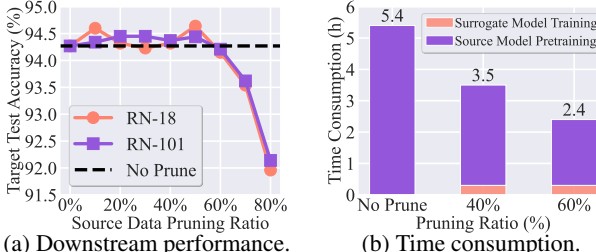

(a) Downstream performance.  (b) Time consumption.

Figure 3: Preliminary studies on the usage of surrogate models for LM. (**a**) The downstream performance (on Flowers102) of using the source model ResNet-101 trained on the pruned ImageNet delivered by LM at different pruning ratios. Here LM is conducted using either RN-101 or a smaller surrogate model ResNet-18. (**b**) Computation time decomposition analysis for obtaining the pretrained model using ResNet-18 as surrogate model with different pruning ratios.

by training the small surrogate model (ResNet-18) for DP is insignificant, compared to the time saved during the pretraining phase of a larger source model (ResNet-101) on the pruned dataset for transfer learning. *Lastly*, pretraining on the subset found by LM can guide the source model towards a flatter region in the downstream loss landscape, (see results in **Fig. A4**). The source model trained on the LM-pruned dataset achieves a higher flatness score than baselines, which aligns with the understanding of transfer learning in [78].

**FM-based DP framework for self-supervised pretraining.** The requirement for *labeled* source data in LM may pose limitations on the application of DP methods, particularly in the context of self-supervised learning (**SSL**)-based pertaining. To address this limitation, we introduce a new method called FM (Feature Mapping). Unlike LM, FM determines the DP scheme using only the feature extractor network $f_\mathcal{S}$, the unlabeled source dataset $\mathcal{D}_\mathcal{S}$, and the target dataset $\mathcal{D}_\mathcal{T}$. This allows us to overcome the dependence on labeled source data, making FM applicable in SSL scenarios. The inspiration for FM is derived from the deep clustering technique [103–105] operating in the representation space, which can generate *pseudo source labels* using *cluster indices* provided by *e.g.*, $K$-means. With the assistance of deep clustering, we can represent the unlabeled source dataset $\mathcal{D}_\mathcal{S}$ as $\mathcal{D}_\mathcal{S} = \{\mathcal{C}_\mathcal{S}^{(1)}, \mathcal{C}_\mathcal{S}^{(2)}, \ldots, \mathcal{C}_\mathcal{S}^{(K)}\}$, where $\mathcal{C}_\mathcal{S}^{(k)}$ denotes the set of source data samples within cluster $k$. Building upon the similar spirit as LM, we propose ranking the importance of pseudo source classes for DP by evaluating the source feature extractor's responsiveness to target data samples. To achieve this objective, we quantify the responsiveness of $f_\mathcal{S}$ for a target sample $\mathbf{t}$ as follows:

$$r(\mathbf{t}) = \arg\min_{k \in [K]} \left\| f_\mathcal{S}(\mathbf{t}) - \mathbb{E}_{\mathbf{x} \in \mathcal{C}_\mathcal{S}^{(k)}}[f_\mathcal{S}(\mathbf{x})] \right\|_2, \tag{1}$$

where $\mathbb{E}_{\mathbf{x} \in \mathcal{C}_\mathcal{S}^{(k)}}[f_\mathcal{S}(\mathbf{x})]$ is the centroid of source data within the cluster $k$, and $r(\mathbf{t})$ is the nearest pseudo label as the responsiveness of $f_\mathcal{S}$ against $\mathbf{t}$. The FM score then integrates (1) with (LM):

$$\mathbf{s}_{\mathrm{FM}}(\mathcal{C}_\mathcal{S}^{(i)}) = \sum_{j=1}^n \mathbb{1}(r(\mathbf{t}_j) = i), \ \text{for } i = 1, 2, \ldots, K, \tag{FM}$$

where different from (LM), $K$ represents the number of pseudo source classes produced by deep clustering, and $r(\mathbf{t}_j)$ corresponds to the source feature extractor's prediction on $\mathbf{t}_j$. It is important to note that the value of $K$ is a free parameter of the deep clustering process. Our empirical study in Fig. A2 shows that FM is quite robust to the choice of $K$ without sacrificing the benefit of DP for

transfer learning compared to using the unpruned source dataset. Lastly, it is worth mentioning that FM can also be applied in the context of supervised pretraining by specifying data labels as clusters.

## 5 Experiments

In this section, we provide a comprehensive set of experiments and analyses to showcase the effectiveness of our proposed methods (LM and FM) in diverse transfer learning scenarios.

### 5.1 Experiment setup

**Datasets and models.** In line with existing transfer learning benchmarks [41, 106], we utilize ImageNet-1K [55] for pretraining and **8** datasets as downstream tasks. These datasets include DTD [107], Flowers102 [102], UCF101 [108], Food101 [109], SUN397 [110], OxfordPets [56], StanfordCars [57], and CIFAR10 [111]. Please refer to Tab. A1 for more details about the datasets. As discussed in Sec. 4, we utilize ResNet-18 (RN-18) [54] as the surrogate source model for pruning source classes. This method significantly reduces the computational cost associated with DP, making the process more efficient. Subsequently, a range of larger models, *e.g.,* ResNet-101 (RN-101) and ViT-B/16 [112], are trained on the (pruned) ImageNet and then finetuned on downstream tasks.

**Baselines, training, and evaluation.** By examining the performance of the existing **8** DP baselines as shown in Fig. 2, our experiments focus on two of the most effective methods: ① GRAND [26] and ② MODERATE [3], together with ③ RANDOM (the random pruning strategy). In Fig. A1, we show more results of the rest DP baselines. Unfortunately, we are unable to include the existing data attribution method [41] as our baseline, as it does not release its pruning results, and we are unable to repeat its experiments due to the need for intensive computations. Unless specified otherwise, we focus on the supervised pretraining setting in the experiments. For self-supervised pretraining, we follow the implementation of MOCOV2 [45]. The finetuning strategies employed include LP (linear probing), which finetunes the classification head with fixed feature extractor, and FF (full finetuning), which finetunes the entire source model. For FM-based DP method, we utilize K-means clustering to group the ImageNet training data points into $K = 2000$ clusters for the computation of (1).

In accordance with the terminology convention in model pruning [87, 113–115], we refer the term '**winning subsets**' to the obtained source subsets that do *not* compromise downstream performance. Among these winning subsets, we identify the one with the highest pruning ratio as the '**best winning subset**'. We then evaluate the performance of DP methods from the following aspects: the downstream performance of the model pretrained on the pruned source dataset obtained by various DP methods, and the pruning ratio of the best winning subset achieved by DP methods, accompanied by the corresponding time saved in the pretraining phase.

### 5.2 Experiment results

**LM/FM improves transfer learning accuracy by identifying 'winning subsets'.** We first showcase the significant improvements achieved by our proposed DP methods (LM and FM) compared to baselines. Our methods successfully identify winning subsets of the ImageNet, yielding transfer accuracy on par or even better than scenarios without pruning.

**Fig. 4** presents the downstream accuracy of transfer learning vs. different pruning ratios. Here DP is performed using the surrogate model (RN-18) on ImageNet for 8 downstream tasks. The source model (RN-101) is then trained on the pruned ImageNet and the transfer learning accuracy is assessed through LP (linear probing) and FF (full finetuning). We also present the downstream performance without pruning the source dataset (No Prune) as a reference for winning subsets. As we can see, both LM and FM significantly outperform the baselines by a substantial margin. Notably, LM and FM consistently identify winning subsets with significantly larger pruning ratios in all settings. This highlights the effectiveness of our proposed methods in achieving substantial dataset pruning without hurting downstream performance. Furthermore, we observe that the downstream performance of LM and FM initially improves and then declines as the pruning ratio increases. This is not surprising, since the initial increase in performance corresponds to the scenario where harmful source classes are removed, consistent with [41]. When the source dataset continues to shrink, the performance inevitably decreases, as a natural consequence of reducing the size of the source dataset. Moreover, in some datasets of small sizes (*e.g.*, OxfordPets), LP achieves performance comparable to FF. This

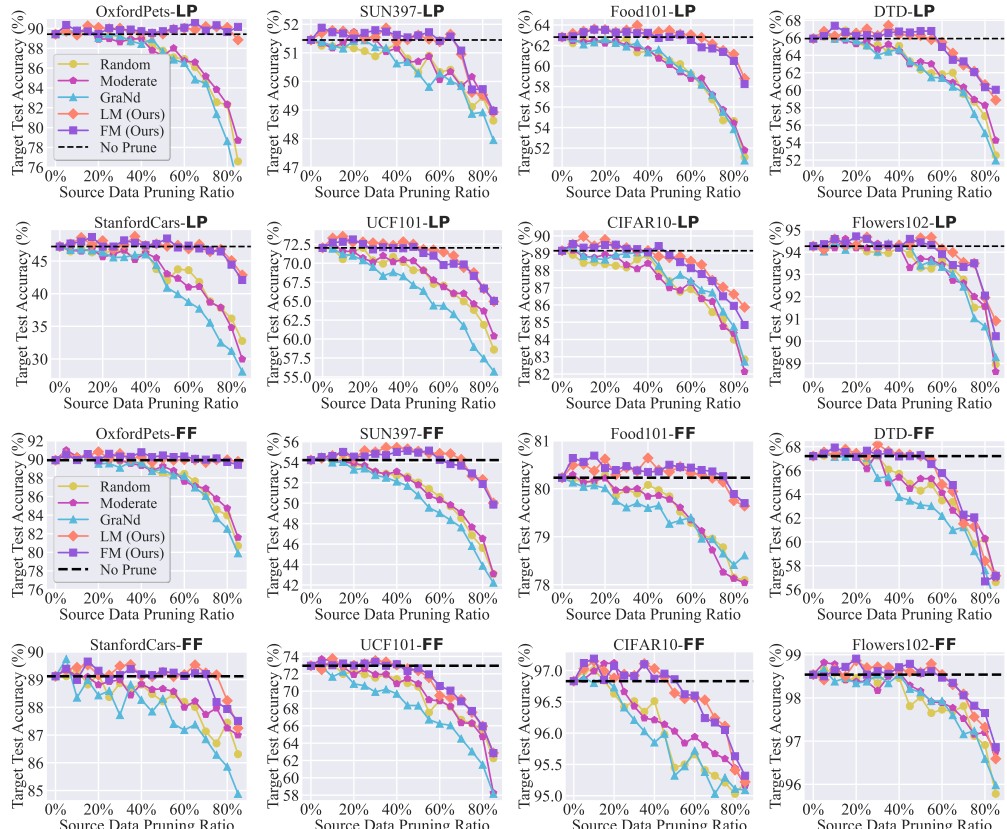

Figure 4: Source dataset pruning trajectory given by downstream testing accuracy (%) vs. source dataset pruning ratio (%) in the supervised pretraining setting. Here the source model RN-101 is trained on each full/pruned source dataset (ImageNet) and finetuned on different downstream tasks through LP and FF. The downstream performance without pruning (No Prune) is marked with the black dashed line. Results are averaged over three independent trials (see exact numbers and variances in Tab. A2).

is attributed to the fact that the performance gap between LP and FF using the large-scale RN-101 model on small-scale datasets tends to diminish.

Table 1: The pruning ratio of the **best winning subsets** obtained by each DP method for different downstream tasks. The other experiment setup is consistent with **Fig. 4**. The achieved largest pruning ratio is highlighted in **bold** in each column. And N/A indicates the scenario where no winning subsets are found.

| Dataset | OxfordPets | | SUN397 | | Food101 | | DTD | | StanfordCars | | UCF101 | | CIFAR10 | | Flowers102 | |
|---|---|---|---|---|---|---|---|---|---|---|---|---|---|---|---|---|
| Finetune method | LP | FF | LP | FF | LP | FF | LP | FF | LP | FF | LP | FF | LP | FF | LP | FF |
| RANDOM | 20% | 20% | 15% | N/A | N/A | 20% | 10% | 20% | N/A | 20% | 20% | 10% | N/A | 15% | 40% | 35% |
| MODERATE | 15% | 20% | 15% | 20% | N/A | 20% | 15% | 30% | N/A | 5% | 5% | 10% | N/A | 20% | 40% | 35% |
| GRAND | 25% | 20% | 10% | 20% | 5% | N/A | 15% | 20% | N/A | 5% | 5% | N/A | 40% | 15% | 45% | 40% |
| LM (ours) | 80% | **80%** | **65%** | 70% | **65%** | 70% | **55%** | 50% | **65%** | **75%** | **45%** | **40%** | 40% | 40% | **55%** | **60%** |
| FM (ours) | **85%** | 75% | **65%** | 60% | 55% | **75%** | **55%** | **50%** | 60% | 70% | **45%** | **40%** | **45%** | **45%** | **55%** | 55% |

**Tab. 1** provides a summary of the pruning ratios achieved by the best winning subsets identified using different DP methods for all 8 downstream datasets. Both LM and FM methods successfully remove more than 45% of the source classes without downstream performance drop. In contrast, all the baselines experience significant performance degradation when the pruning ratio exceeds 40%.

Table 2: The downstream performance with different source data pruning ratios in the SSL pretraining setting. A randomly initialized RN-101 is self-supervised pretrained using MOCO V2 on each full/pruned source dataset and finetuned on the downstream task through LP. The best result in each pruning ratio is marked in **bold** and the performance surpassing the unpruned setting (pruning ratio 0%) is highlighted in cyan .

| Dataset | OxfordPets | | | | | SUN397 | | | | | Flowers102 | | | | |
|---|---|---|---|---|---|---|---|---|---|---|---|---|---|---|---|
| Pruning Ratio | 0% | 50% | 60% | 70% | 80% | 0% | 50% | 60% | 70% | 80% | 0% | 50% | 60% | 70% | 80% |
| RANDOM | | 62.32 | 61.27 | 59.09 | 53.75 | | 45.63 | 45.08 | 43.54 | 39.81 | | 82.23 | 82.60 | 81.03 | 80.02 |
| MODERATE | 69.26 | 63.37 | 62.45 | 63.31 | 57.42 | 47.36 | 45.73 | 45.14 | 44.23 | 40.82 | 85.17 | 82.45 | 81.45 | 81.69 | 81.32 |
| GRAND | | 64.42 | 63.34 | 61.14 | 56.42 | | 45.72 | 45.58 | 45.24 | 41.72 | | 82.85 | 82.44 | 82.14 | 81.73 |
| FM (ours) | | **69.92** | **69.99** | **70.29** | **70.21** | | **48.46** | **48.58** | **47.90** | **46.00** | | **85.22** | **85.42** | **84.37** | **84.61** |

**Tab. 2** highlights the effectiveness of FM-based DP in the self-supervised pretraining setup for three representative downstream tasks. As we can see, the transfer learning accuracy achieved by using FM consistently outperforms baselines in the self-supervised pretraining paradigm. FM can identify winning subsets for transfer learning even in the challenging regime of large pruning ratios, ranging from 50% to 80%. For instance, in the case of SUN397, FM-based winning subsets achieves a pruning ratio up to 70%, and for Flowers102 the maximum pruning ratio is 60%. These pruning merits align with the findings for FM in the supervised pretraining setup, as illustrated in Tab. 1.

**DP enhances the efficiency of source pretraining. Tab. 3** displays the computation time required to obtain the pretrained source model using LM at different pruning ratios. The reported time consumption includes the entire pipeline, encompassing surrogate model training (RN-18), DP process, and source model training (RN-101) on the pruned ImageNet dataset.

Table 3: Time consumption of LM/FM in Fig.4 to obtain the pretrained model. The reported time consumption covers surrogate model (RN-18) training, LM/FM dataset pruning, and source model pretraining (RN-101).

| Pruning Ratio | 0% | 20% | 40% | 60% | 80% |
|---|---|---|---|---|---|
| Time Consumption (h) | 5.4 | 4.6 (15%↓) | 3.5 (35%↓) | 2.4 (56%↓) | 1.3 (76%↓) |

The runtime cost of the conventional transfer learning on the full ImageNet dataset for RN-101 is also listed as a reference. As we can see, DP enjoys high efficiency merit of source training. Taking the 5.4 hours required for source training on the full ImageNet dataset as a reference, LM-enabled 20% pruned ImageNet achieves a 15% reduction in training time. Moreover, the efficiency advantage increases to 76% when the pruning ratio reaches 80% and these computational benefits do not sacrifice transfer learning accuracy at all.

Next, we compare the efficiency of our methods with the state-of-the-art dataset pruning baselines in Tab. 4, including GRAND, MODERATE, and the brute-force method proposed in [41]. We showcase the efficiency comparison with a pruning ratio of 60%. As we can see, our method

Table 4: Time consumption comparison with a pruning ratio of 60% of different dataset pruning methods. Other settings follow Fig. 4.

| Method | MODERATE | GRAND | BRUTE-FORCE [41] | ours |
|---|---|---|---|---|
| Time Consumption (h) | 7.6 | 18.4 | > 500 | 2.4 |

(even using a surrogate model) achieves a substantial computation efficiency improvement over other methods. In particular, [41] involves training thousands of source models. Thus, its computational cost is typically unaffordable in experiments.

**DP enables efficient adversarial pretraining. Tab. 5** showcases the improvement in transfer learning accuracy and efficiency achieved by our proposed LP-based DP method when incorporating *adversarial training* (**AT**) [116] on either the full or pruned ImageNet dataset. This transfer learning setup is motivated by the findings in [58], showing that enhancing the robustness of the source model against adversarial attacks through AT can improve transfer learning accuracy in both LP and FF-based finetuning scenarios. We then report downstream accuracies

Table 5: Downstream performance of models pretrained on full/pruned source dataset (DENSE/LM) w/o adversarial pretraining (Adv). For DENSE-ADV and LM-ADV, 3-step adversarial training [116] is used for pretraining. Pruning ratio, downstream test accuracy (Acc.), and time consumptions for obtaining pretrained models are reported.

| Method | | SUN397 | | | | DTD | | |
|---|---|---|---|---|---|---|---|---|
| | Pruning Ratio | Acc.(%) | | Time (h) | Pruning Ratio | Acc.(%) | | Time (h) |
| | | LP | FF | | | LP | FF | |
| DENSE | N/A | 51.45 | 54.21 | 5.4 | N/A | 65.91 | 67.21 | 5.4 |
| DENSE-ADV | N/A | 52.97 | 55.67 | 13.7 | N/A | 67.23 | 68.92 | 13.7 |
| LM | 70 | 50.95 | 54.28 | 1.9 | 50 | 66.25 | 67.22 | 2.9 |
| LM-ADV | 70 | 52.07 | 55.49 | 4.2 | 50 | 67.02 | 68.54 | 6.7 |

using LP and FF on two specific downstream datasets: SUN397 and DTD, which are intentionally chosen due to the large room for improvement in transfer learning accuracy, as shown in Fig. 4. We also determine the pruning ratios for LM by selecting those that led to the best winning subsets. Our experimental results demonstrate that employing AT on both the unpruned and pruned source datasets can improve transfer learning accuracy. Specifically, we refer to AT on the original unpruned ImageNet dataset as DENSE-AT, and AT on the LM-pruned ImageNet dataset as LM-AT. One notable advantage of integrating LM into AT is the significant improvement in computation efficiency. A key highlight of this approach is that LM-AT achieves a similar computation time as the standard source training on ImageNet (DENSE), while exhibiting almost no accuracy drop compared to DENSE-AT. This observation demonstrates the potential to accelerate AT through DP.

**Robustness against the surrogate model size.** To explore the sensitivity of our proposed method to the surrogate model's size and accuracy, we show the transfer learning performance on the task Oxford-Pets against different surrogate model sizes in **Fig. 5**. It is evident that even though the performance of the surrogate model on the source dataset (ImageNet) decreases, the downstream performance of RN-101 pretrained on the LM-based pruned source subsets remains relatively stable. Further, **Tab. A6** compares the class indices selected by the corresponding surrogate models. Interestingly, the most relevant source class selections exhibit a high level of agreement across surrogate models of differing sizes.

LM shows robust behavior with different surrogate models, even if they yield different performance on the source dataset due to their different model capacities. For example, RN-32s only achieves a test accuracy of $40.77\%$ on the source dataset, but it manages to achieve the same maximum pruning ratio of the winning subsets as RN-18. This underscores that LM can accommodate a small surrogate model of $1\%$ the size of the pretrained model (Rn-32s vs. RN-101). Moreover, it reveals that the surrogate model does not need to attain exceptional performance in the source domain for dataset pruning in transfer learning.

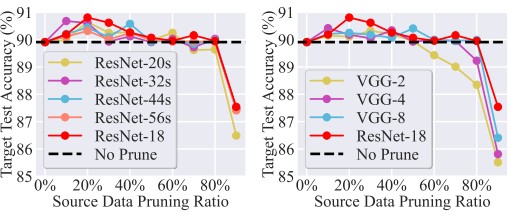

(a) ResNets family    (b) VGG family

Figure 5: Source dataset pruning trajectory given the downstream task OxfordPets using different surrogate models. Other settings follow Fig. 4.

Table 6: Downstream performance of different DP methods using ViT-B/16 as the source model and FF as the finetuning method. Other settings are consistent with Fig. 4. The best performance at each pruning ratio is marked in **bold**, and performance exceeding the unpruned setting ($0\%$ pruning ratio) is highlighted in `cyan`.

| Dataset | | OxfordPets | | | | | SUN397 | | | | | Flowers102 | | | |
|---|---|---|---|---|---|---|---|---|---|---|---|---|---|---|---|
| Pruning Ratio | 0% | 20% | 40% | 60% | 80% | 0% | 20% | 40% | 60% | 80% | 0% | 20% | 40% | 60% | 80% |
| RANDOM | | 85.63 | 83.72 | 75.94 | 69.83 | | 43.74 | 41.18 | 36.72 | 29.41 | | 90.25 | 89.38 | 87.47 | 84.44 |
| MODERATE | 85.75 | 85.39 | 83.97 | 82.39 | 78.32 | 43.97 | 43.97 | 41.82 | 38.89 | 31.15 | 90.95 | 90.86 | 89.77 | 88.34 | 86.69 |
| GRAND | | 85.77 | 83.88 | 82.35 | 79.12 | | 43.32 | 40.93 | 38.52 | 31.00 | | 90.75 | 89.89 | 88.54 | 85.39 |
| LM | | **86.15** | **85.72** | **84.77** | **80.64** | | **44.24** | **42.79** | **39.59** | **32.35** | | **91.19** | **89.97** | **89.07** | **87.23** |

**DP for ViT.** In **Tab. 6**, we present the consistent transfer learning improvement achieved by LM for Vision Transformer (ViT) pretraining. Specifically, we utilize ViT-B/16 as the source model, RN-18 as the surrogate model for DP, and use FF on the source model for the downstream tasks studied in Tab. 2. As we can see, LM yields a better transfer learning accuracy than baselines at different pruning ratios, consistent with the previous results on RN-101. This observation highlights the model-agnostic advantage of our proposed method in transfer learning. We also observe the pruning ratio of the best winning subset achieved for ViT is smaller compared to ResNet. This discrepancy is possibly due to the higher data requirements of ViT during pretraining [112].

**Additional results.** We conduct ablation studies on the cluster number for FM in Fig. A2 and the results of selecting data proposed by LM in reverse order in Fig. A3. We shows FM can be applied to more advanced SSL framework in Tab. A3. We also investigate the performance of our proposed method in the multi-task setting in Fig. A5, and the scenarios with data biases in Tab. A4. We further examine our method on the few-shot benchmark in Tab. A5. We also present feature distribution analysis in Fig. A6 and image examples for the top classes selected by LM/FM in Fig. A7.

**Limitations.** Despite the remarkable empirical results above, we admit LM/FM mainly caters to exploring the source data influence in a specific downstream task, and the largest pruning ratio is reduced as more downstream tasks are considered (see Fig. A5), which we consider as a limitation. Meanwhile, we acknowledge that the promising performance achieved by our method still lacks rigorous theoretical analysis, despite the feature distribution and model flatness analysis, which we believe will be a challenging but exciting future work on DP for transfer learning.

# 6 Conclusion

In this paper, we first formally define and investigate the problem of DP (dataset pruning) for transfer learning. Recognizing the ineffectiveness of conventional DP methods in the transfer learning setting, we proposed two novel and efficient DP techniques, label mapping and feature mapping. Extensive experiments demonstrate that both methods can effectively prune a large ratio of source dataset and substantially reduce the pretraining computational costs without sacrificing the downstream performance in various transfer learning settings.

# Acknowledgement

The work of Y. Zhang, Y. Zhang, A. Chen, J. Jia, J. Liu, S. Chang, and S. Liu was supported by the Cisco Research Award and partially supported by the NSF Grant IIS-2207052. The work of M. Hong was supported by NSF grants CIF-1910385 and EPCN-2311007.

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

# Appendix

## A  Experiment Settings

### A.1  Training Settings

**General setups for DP (dataset pruning).**  Regardless of the choice of the downstream tasks and DP methods, each DP process will go through a three-stage paradigm, namely ❶ source dataset pruning, ❷ pretraining based on the pruned source dataset, and ❸ finetuning on the target dataset using either LP (linear probing) or FF (full finetuning). Regarding baseline methods for source dataset pruning, we strictly follow the baselines' settings provided in the literature. As stated before, we utilize a small surrogate model (ResNet-18) pretrained on the full ImageNet to conduct DP. The proportion of pruned class numbers to the total source class numbers determines pruning ratios for our methods: LM (label mapping) and FM (feature mapping). To ensure a fair comparison with other non-class-wise baselines (RANDOM, GRAND, MODERATE), the same pruning ratio is applied to the total number of training data points.

**Pretraining setups.**  We keep our pretraining procedure consistent for all models and strictly follow the settings released by the existing work [41]. All the models are trained from scratch using Stochastic Gradient Descent (SGD) to minimize the standard cross-entropy loss in a multi-class classification setting. We use a batch size of 1024, momentum of 0.9, and a weight decay of $5 \times 10^{-4}$. The training utilizes a cyclic learning rate schedule, which begins with an initial learning rate of 0.5 and peaks at the second epoch. During training, we incorporate data augmentations such as random resized cropping and random horizontal flipping.

**Downstream task finetuning settings.**  Our approach to finetuning the pretrained model on downstream tasks involves using LP and FF. Details of the downstream datasets and the training configurations are presented in Tab. A1, following [98]. For LP, we employ the Adam optimizer, a multi-step decaying scheduler, and an initial learning rate of 0.1 across 50 total training epochs. As for FF, we utilize the Adam optimizer over 200 epochs with a cosine-annealing scheduler, an initial learning rate of 0.01, and a weight decay of $5 \times 10^{-4}$. All finetuning experiments employ a batch size of 256 and standard data augmentations, such as random resized cropping and horizontal flipping.

**Training settings for SSL (self-supervised learning) and ViT.**  Our SSL training settings follow the configurations provided by MOCOV2. Details of the pretraining and finetuning stages can be accessed at https://github.com/facebookresearch/moco. For the training of ViTs, we rely on the setting released in the original ViT paper [112] (see ViT/B in Table 3).

Table A1: Dataset attributes and training configurations of 8 downstream image classification datasets considered in this work.

| Dataset | Train Size | Test Size | Class Number | Batch Size | Rescaled Resolution |
|---|---|---|---|---|---|
| Flowers102 | 4093 | 2463 | 102 | 256 | 224×224 |
| DTD | 2820 | 1692 | 47 | 256 | 224×224 |
| UCF101 | 7639 | 3783 | 101 | 256 | 224×224 |
| Food101 | 50500 | 30300 | 101 | 256 | 224×224 |
| OxfordPets | 2944 | 3669 | 37 | 256 | 224×224 |
| StanfordCars | 6509 | 8041 | 196 | 256 | 224×224 |
| SUN397 | 15888 | 19850 | 397 | 256 | 224×224 |
| CIFAR10 | 50000 | 10000 | 10 | 256 | 160×160 |

## B  Additional Results

**Expanded performance evaluation of in-domain DP methods across all downstream datasets.** **Fig. A1** expands on the performance comparisons in **Fig. 2**, providing a more thorough evaluation of various in-domain DP methods on all eight downstream datasets. The trends observed are consistent with those in **Fig. 2**: Random pruning shows a strong baseline method in DP for transfer learning compared to other state-of-the-art DP methods designed for non-transfer learning. This observation prompts us to explore more effective dataset pruning strategies for transfer learning. MODERATE and

GRAND are also demonstrating strong baselines, motivating us to choose them as the default DP baselines in Section 5.

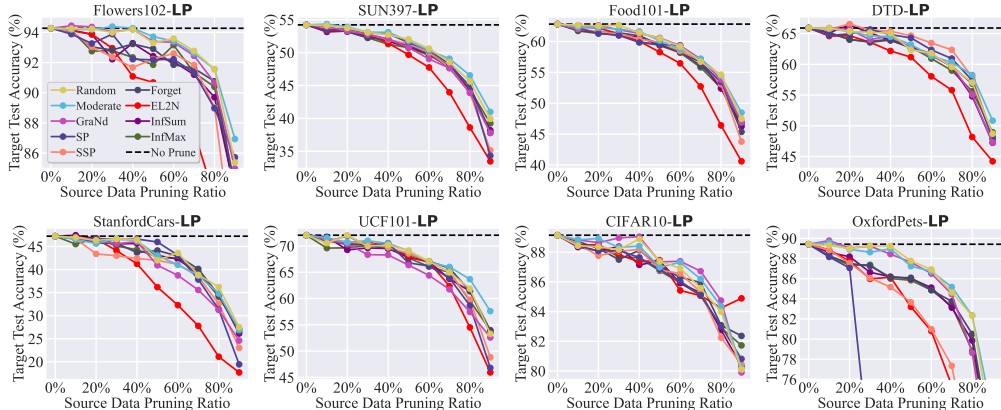

Figure A1: Extended performance comparison of different in-domain DP methods for transfer learning across all downstream tasks. Experiment settings are consistent with Fig. 4.

**The main results reported in Fig. 4 are stable: analysis with detailed numerical results and standard deviations.** In **Tab. A2**, we provide the exact numerical results used to generate **Fig. 4**. These numbers give a more granular view of the performance comparisons. The results, calculated over three independent trials, show that the magnitude of the standard deviations is quite small relative to the mean values. This indicates that the trends and conclusions drawn from **Fig. 4** are generally valid and not significantly affected by trial variations. To maintain readability and clarity, we choose not to include these minor standard deviations in **Fig. 4**.

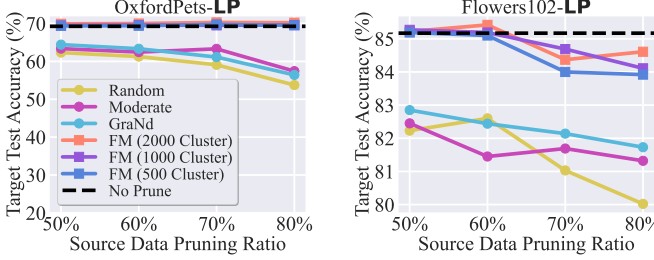

Figure A2: Ablation study on the sensitivity of FM against the choice of cluster numbers. For FM, the original dataset is pre-clustered into $K$ clusters, with $K = 2000$ as the default setting in this study. Here, various $K$ values (500, 1000, and 2000) are studied in this experiment. Other settings are aligned with Tab. 2.

**FM performance is robust to the choice of cluster number.** The results presented in **Fig. A2** provide insights into the sensitivity of FM performance to variations of the cluster number ($K$). This extended study from Tab. 2 shows that FM performance remains robust as the cluster number varies from 500 to 2000, preserving the benefits of DP for transfer learning without much performance degradation. This implies that FM can provide reliable and stable results even when the hyperparameter settings are not optimal.

**LM identifies the most influential source classes.** To validate that the classes with the highest LM scores are the most influential, we present the pruning trajectory in **Fig. A3**, where pruning is executed in the reverse order of the class scores, different from the proposed DP implementation. That is, classes with the smallest scores are retained, while the ones with the highest scores are pruned. Remarkably, even a slight pruning ratio (such as 10%) leads to a significant degradation in downstream performance. This evidence underscores the benefits of source classes with high LM scores in promoting the downstream performance.

**FM can be smoothly applied to MoCov3.** We conducted experiments to illustrate the effectiveness of our proposed method (FM) when applied to the more recent SSL framework MoCov3 [117] using

Table A2: Exact numbers and standard deviations for Fig. 4.

| Method | | | | | | | | | Pruning Ratio | | | | | | | | |
|---|---|---|---|---|---|---|---|---|---|---|---|---|---|---|---|---|---|
| | 5% | 10% | 15% | 20% | 25% | 30% | 35% | 40% | 45% | 50% | 55% | 60% | 65% | 70% | 75% | 80% | 85% |
| **OxfordPets - LP** | | | | | | | | | | | | | | | | | |
| RANDOM | 89.81±0.22 | 89.32±0.18 | 89.92±0.12 | 88.99±0.34 | 89.26±0.13 | 89.26±0.11 | 88.66±0.38 | 89.23±0.28 | 88.23±0.07 | 87.76±0.05 | 87.08±0.27 | 86.92±0.23 | 86.10±0.03 | 84.63±0.31 | 82.56±0.09 | 82.31±0.2 | 76.59±0.35 |
| MODERATE | 89.67±0.29 | 89.59±0.29 | 89.86±0.15 | 88.91±0.14 | 88.96±0.10 | 88.63±0.25 | 88.93±0.32 | 88.96±0.05 | 87.79±0.37 | 87.24±0.17 | 88.01±0.23 | 86.73±0.02 | 86.59±0.04 | 85.17±0.08 | 83.84±0.04 | 82.34±0.33 | 78.71±0.27 |
| GRAND | 89.70±0.20 | 89.78±0.16 | 89.78±0.24 | 89.04±0.38 | 89.40±0.11 | 89.23±0.07 | 88.72±0.04 | 88.44±0.39 | 88.93±0.28 | 87.63±0.20 | 86.86±0.29 | 86.51±0.17 | 84.85±0.02 | 84.46±0.14 | 81.36±0.33 | 78.63±0.13 | 74.19±0.08 |
| LM | 89.72±0.1 | 89.34±0.04 | 90.27±0.37 | 90.11±0.27 | 89.97±0.38 | 90.02±0.19 | 89.72±0.15 | 89.62±0.24 | 89.46±0.16 | 89.45±0.06 | 89.92±0.11 | 90.02±0.27 | 89.89±0.18 | 90.19±0.14 | 90.16±0.38 | 90.11±0.03 | 88.85±0.37 |
| FM | 90.35±0.32 | 89.78±0.07 | 89.70±0.24 | 89.45±0.29 | 90.02±0.22 | 89.83±0.37 | 89.89±0.03 | 89.72±0.27 | 89.56±0.06 | 89.72±0.19 | 89.94±0.11 | 90.08±0.36 | 90.62±0.03 | 89.78±0.10 | 90.32±0.38 | 89.64±0.15 | 90.16±0.07 |
| **SUN397 - LP** | | | | | | | | | | | | | | | | | |
| RANDOM | 51.25±0.14 | 51.23±0.37 | 51.23±0.27 | 51.16±0.32 | 51.06±0.21 | 50.88±0.08 | 51.11±0.34 | 51.32±0.28 | 50.80±0.19 | 50.31±0.27 | 50.90±0.36 | 50.33±0.17 | 50.41±0.07 | 49.88±0.24 | 49.11±0.10 | 49.45±0.14 | 48.63±0.31 |
| MODERATE | 51.34±0.10 | 51.17±0.35 | 51.36±0.38 | 51.73±0.07 | 51.45±0.20 | 51.48±0.36 | 51.04±0.12 | 51.16±0.15 | 50.68±0.14 | 50.70±0.21 | 50.87±0.27 | 50.06±0.36 | 50.34±0.38 | 49.83±0.12 | 50.16±0.34 | 49.51±0.25 | 49.00±0.20 |
| GRAND | 51.38±0.30 | 51.25±0.24 | 51.15±0.33 | 51.42±0.14 | 51.44±0.38 | 51.20±0.24 | 51.17±0.13 | 50.63±0.42 | 50.68±0.29 | 50.28±0.36 | 49.81±0.38 | 50.28±0.10 | 50.02±0.28 | 49.84±0.12 | 48.87±0.21 | 48.92±0.11 | 47.95±0.27 |
| LM | 51.70±0.18 | 51.78±0.26 | 51.76±0.20 | 51.67±0.35 | 51.44±0.38 | 51.55±0.28 | 51.87±0.10 | 51.40±0.16 | 51.46±0.32 | 51.48±0.23 | 51.46±0.32 | 51.65±0.18 | 51.65±0.18 | 50.95±0.33 | 49.62±0.16 | 49.53±0.21 | 48.93±0.27 |
| FM | 51.88±0.26 | 51.72±0.31 | 51.68±0.18 | 51.50±0.14 | 51.70±0.37 | 51.79±0.30 | 51.74±0.24 | 51.63±0.38 | 51.54±0.33 | 51.61±0.28 | 51.72±0.27 | 51.42±0.32 | 51.57±0.23 | 51.08±0.22 | 49.72±0.18 | 49.73±0.21 | 48.97±0.35 |
| **Food101 - LP** | | | | | | | | | | | | | | | | | |
| RANDOM | 62.22±0.30 | 62.64±0.21 | 62.32±0.18 | 62.53±0.33 | 62.23±0.17 | 62.65±0.27 | 61.33±0.31 | 61.42±0.39 | 60.91±0.26 | 60.50±0.21 | 60.19±0.34 | 59.20±0.20 | 58.15±0.32 | 56.74±0.27 | 54.70±0.19 | 54.64±0.24 | 51.12±0.22 |
| MODERATE | 62.47±0.19 | 62.34±0.28 | 62.49±0.27 | 62.22±0.16 | 62.45±0.36 | 62.00±0.24 | 61.76±0.30 | 61.63±0.29 | 60.75±0.16 | 60.13±0.26 | 59.42±0.24 | 58.78±0.35 | 58.80±0.30 | 57.19±0.25 | 55.73±0.37 | 54.43±0.24 | 51.79±0.16 |
| GRAND | 62.70±0.18 | 62.10±0.24 | 62.30±0.33 | 62.61±0.22 | 62.25±0.27 | 61.90±0.24 | 62.27±0.31 | 61.32±0.34 | 61.60±0.22 | 60.58±0.24 | 59.72±0.25 | 59.32±0.33 | 58.19±0.27 | 57.17±0.21 | 55.54±0.19 | 53.84±0.30 | 50.79±0.28 |
| LM | 63.17±0.16 | 63.39±0.27 | 63.60±0.18 | 63.55±0.32 | 63.30±0.16 | 63.50±0.33 | 63.97±0.23 | 63.08±0.20 | 63.34±0.25 | 63.47±0.33 | 63.09±0.23 | 63.05±0.36 | 62.78±0.26 | 62.14±0.30 | 61.51±0.33 | 61.15±0.22 | 58.78±0.30 |
| FM | 62.89±0.24 | 63.29±0.24 | 63.55±0.16 | 63.53±0.37 | 63.26±0.30 | 63.34±0.27 | 63.20±0.35 | 63.24±0.22 | 62.73±0.32 | 62.84±0.24 | 63.06±0.23 | 62.55±0.16 | 61.82±0.35 | 61.71±0.32 | 61.38±0.24 | 60.50±0.36 | 58.24±0.32 |
| **DTD - LP** | | | | | | | | | | | | | | | | | |
| RANDOM | 66.13±0.20 | 65.90±0.17 | 66.84±0.21 | 65.37±0.23 | 65.07±0.30 | 65.19±0.19 | 64.42±0.22 | 65.08±0.18 | 63.30±0.25 | 62.35±0.32 | 61.95±0.29 | 61.82±0.24 | 62.00±0.28 | 59.57±0.31 | 58.69±0.34 | 57.03±0.33 | 52.54±0.26 |
| MODERATE | 66.73±0.18 | 65.84±0.30 | 65.72±0.16 | 65.31±0.33 | 65.60±0.22 | 64.66±0.27 | 65.25±0.16 | 64.48±0.35 | 63.00±0.22 | 63.24±0.36 | 63.00±0.33 | 61.35±0.24 | 60.87±0.25 | 60.34±0.16 | 58.92±0.34 | 58.27±0.28 | 54.26±0.24 |
| GRAND | 66.67±0.24 | 65.90±0.22 | 65.96±0.17 | 65.66±0.26 | 65.13±0.27 | 64.01±0.35 | 64.42±0.16 | 64.66±0.22 | 63.30±0.28 | 62.65±0.16 | 61.47±0.23 | 61.41±0.29 | 60.40±0.19 | 59.75±0.35 | 57.27±0.16 | 55.08±0.28 | 51.95±0.26 |
| LM | 66.73±0.17 | 66.21±0.21 | 66.43±0.22 | 66.54±0.25 | 67.02±0.33 | 66.25±0.28 | 67.43±0.22 | 66.38±0.24 | 66.33±0.14 | 66.25±0.21 | 65.80±0.22 | 65.48±0.16 | 64.24±0.32 | 62.88±0.35 | 62.17±0.26 | 60.64±0.28 | 58.87±0.32 |
| FM | 66.19±0.23 | 67.38±0.32 | 66.67±0.30 | 66.22±0.16 | 66.31±0.22 | 66.13±0.19 | 66.55±0.23 | 66.72±0.17 | 66.61±0.27 | 66.77±0.33 | 66.78±0.22 | 64.95±0.29 | 63.48±0.30 | 63.30±0.35 | 62.07±0.26 | 60.34±0.18 | 60.05±0.23 |
| **StanfordCars - LP** | | | | | | | | | | | | | | | | | |
| RANDOM | 46.47±0.23 | 46.95±0.20 | 46.32±0.21 | 46.33±0.19 | 46.49±0.32 | 46.63±0.28 | 45.39±0.30 | 46.46±0.18 | 43.76±0.25 | 42.00±0.28 | 43.74±0.33 | 43.61±0.22 | 41.80±0.35 | 38.84±0.29 | 37.81±0.30 | 36.20±0.31 | 32.76±0.28 |
| MODERATE | 46.66±0.16 | 46.51±0.24 | 46.85±0.30 | 45.64±0.16 | 46.03±0.21 | 46.67±0.22 | 45.21±0.17 | 46.23±0.32 | 45.47±0.29 | 43.03±0.33 | 42.35±0.30 | 41.00±0.28 | 41.09±0.24 | 38.69±0.26 | 37.89±0.33 | 34.81±0.22 | 29.97±0.32 |
| GRAND | 46.70±0.27 | 46.77±0.19 | 46.62±0.20 | 46.33±0.30 | 45.55±0.16 | 45.59±0.33 | 45.94±0.22 | 46.02±0.26 | 44.40±0.24 | 40.92±0.30 | 39.93±0.31 | 38.73±0.19 | 37.71±0.28 | 35.56±0.34 | 32.51±0.16 | 31.23±0.28 | 28.06±0.26 |
| LM | 47.60±0.28 | 48.60±0.24 | 47.52±0.30 | 48.12±0.33 | 46.81±0.14 | 47.49±0.22 | 48.78±0.27 | 47.36±0.23 | 47.63±0.26 | 47.27±0.19 | 47.42±0.30 | 46.88±0.24 | 47.56±0.26 | 46.45±0.33 | 46.80±0.18 | 45.07±0.25 | 42.87±0.32 |
| FM | 47.69±0.22 | 47.98±0.16 | 48.69±0.33 | 47.21±0.19 | 47.23±0.32 | 48.17±0.18 | 47.80±0.30 | 47.41±0.28 | 47.63±0.26 | 48.46±0.21 | 47.08±0.30 | 47.37±0.16 | 46.93±0.31 | 46.64±0.33 | 46.47±0.22 | 44.45±0.24 | 42.10±0.28 |
| **UCF101 - LP** | | | | | | | | | | | | | | | | | |
| RANDOM | 71.96±0.31 | 70.53±0.28 | 71.53±0.33 | 72.03±0.30 | 70.37±0.24 | 69.94±0.31 | 70.84±0.32 | 70.16±0.27 | 69.07±0.29 | 69.07±0.28 | 67.27±0.25 | 67.06±0.24 | 65.98±0.26 | 64.97±0.30 | 63.84±0.22 | 61.88±0.23 | 58.60±0.32 |
| MODERATE | 72.72±0.24 | 71.56±0.29 | 71.58±0.28 | 70.68±0.33 | 70.13±0.25 | 70.98±0.27 | 70.18±0.29 | 70.47±0.26 | 70.31±0.24 | 69.05±0.24 | 67.78±0.31 | 66.88±0.28 | 66.03±0.29 | 66.01±0.28 | 64.63±0.27 | 63.68±0.33 | 60.38±0.24 |
| GRAND | 71.93±0.22 | 71.19±0.25 | 71.00±0.33 | 70.39±0.29 | 69.52±0.24 | 68.33±0.30 | 68.81±0.32 | 68.28±0.22 | 67.14±0.26 | 66.35±0.31 | 64.42±0.33 | 64.39±0.30 | 63.28±0.31 | 61.75±0.33 | 58.95±0.26 | 57.44±0.33 | 55.70±0.28 |
| LM | 73.32±0.38 | 73.54±0.29 | 72.77±0.25 | 72.82±0.30 | 72.69±0.29 | 72.58±0.24 | 72.35±0.30 | 72.83±0.29 | 72.51±0.27 | 71.82±0.32 | 71.76±0.31 | 71.48±0.33 | 70.98±0.24 | 69.44±0.28 | 68.75±0.26 | 66.64±0.29 | 64.95±0.30 |
| FM | 72.72±0.32 | 72.83±0.27 | 73.14±0.30 | 72.53±0.32 | 71.82±0.29 | 72.11±0.26 | 72.00±0.31 | 71.97±0.29 | 72.16±0.28 | 71.35±0.33 | 71.00±0.24 | 69.76±0.31 | 69.94±0.33 | 69.84±0.30 | 68.33±0.28 | 66.67±0.29 | 65.03±0.33 |
| **CIFAR10 - LP** | | | | | | | | | | | | | | | | | |
| RANDOM | 88.9±0.24 | 88.43±0.28 | 88.46±0.27 | 88.36±0.29 | 88.27±0.31 | 88.17±0.33 | 88.64±0.27 | 88.88±0.27 | 88.01±0.30 | 87.38±0.24 | 86.77±0.28 | 86.91±0.31 | 86.38±0.24 | 85.58±0.32 | 85.18±0.27 | 83.99±0.24 | 82.85±0.33 |
| MODERATE | 89.17±0.30 | 88.83±0.32 | 88.74±0.26 | 88.92±0.33 | 88.82±0.27 | 88.32±0.28 | 88.09±0.26 | 88.41±0.24 | 87.63±0.26 | 86.99±0.27 | 86.85±0.24 | 87.31±0.32 | 86.36±0.33 | 86.20±0.26 | 84.75±0.30 | 84.35±0.31 | 82.14±0.31 |
| GRAND | 89.35±0.29 | 88.83±0.24 | 88.69±0.27 | 88.63±0.28 | 88.91±0.30 | 88.96±0.26 | 88.96±0.26 | 89.04±0.29 | 88.27±0.32 | 87.34±0.27 | 87.75±0.24 | 87.39±0.31 | 86.86±0.27 | 86.72±0.33 | 85.61±0.26 | 84.74±0.31 | 82.72±0.32 |
| LM | 89.51±0.31 | 89.97±0.33 | 89.42±0.24 | 88.79±0.27 | 89.42±0.26 | 89.31±0.32 | 89.14±0.30 | 89.15±0.31 | 88.81±0.33 | 88.74±0.30 | 88.81±0.27 | 88.54±0.26 | 88.33±0.32 | 87.43±0.29 | 87.03±0.30 | 86.62±0.27 | 85.87±0.31 |
| FM | 89.53±0.28 | 89.32±0.30 | 89.32±0.31 | 89.47±0.29 | 89.49±0.32 | 89.25±0.31 | 89.23±0.30 | 89.41±0.33 | 88.89±0.26 | 88.13±0.30 | 88.48±0.29 | 88.13±0.30 | 87.79±0.27 | 87.39±0.33 | 86.50±0.30 | 85.95±0.32 | 84.84±0.27 |
| **Flowers102 - LP** | | | | | | | | | | | | | | | | | |
| RANDOM | 94.14±0.28 | 94.25±0.28 | 94.32±0.27 | 94.23±0.29 | 94.03±0.31 | 93.99±0.33 | 93.96±0.26 | 94.20±0.27 | 93.91±0.30 | 93.30±0.24 | 93.26±0.28 | 93.59±0.31 | 93.26±0.32 | 92.77±0.32 | 91.51±0.27 | 91.56±0.26 | 88.96±0.33 |
| MODERATE | 94.24±0.30 | 94.19±0.32 | 94.6±0.26 | 94.19±0.33 | 94.19±0.27 | 94.38±0.28 | 94.23±0.29 | 94.28±0.24 | 93.3±0.26 | 93.71±0.27 | 93.67±0.29 | 93.42±0.32 | 92.73±0.33 | 92.61±0.26 | 92±0.30 | 91.56±0.26 | 88.63±0.31 |
| GRAND | 94.03±0.29 | 94.44±0.24 | 94.11±0.27 | 94.36±0.28 | 94.68±0.30 | 94.03±0.26 | 94.4±0.33 | 94.2±0.29 | 94.28±0.32 | 93.4±0.27 | 93.59±0.31 | 93.34±0.31 | 93.02±0.27 | 92.45±0.33 | 91.03±0.26 | 90.66±0.31 | 89.28±0.32 |
| LM | 94.11±0.31 | 94.60±0.33 | 94.31±0.26 | 94.32±0.30 | 94.68±0.26 | 94.23±0.32 | 94.29±0.27 | 94.32±0.26 | 94.32±0.32 | 94.64±0.31 | 94.67±0.30 | 94.15±0.24 | 93.99±0.32 | 93.54±0.31 | 93.5±0.26 | 91.96±0.29 | 90.91±0.30 |
| FM | 94.36±0.26 | 94.59±0.33 | 94.40±0.24 | 94.71±0.32 | 94.60±0.31 | 94.28±0.33 | 94.34±0.26 | 94.21±0.31 | 94.60±0.30 | 94.44±0.33 | 94.32±0.29 | 93.91±0.26 | 93.42±0.31 | 93.34±0.24 | 93.50±0.30 | 92.05±0.27 | 90.22±0.26 |

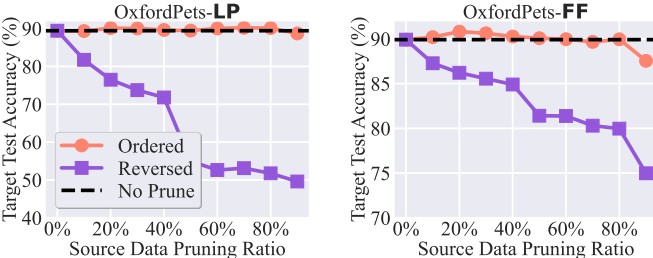

Figure A3: Performance comparison on the downstream performance of LM with the "Ordered" and the "Reversed" pruning order. Here, the "Ordered" strategy retains source classes with the highest scores, while the "Reversed" order prunes these classes first. Other settings are aligned with Fig. 4.

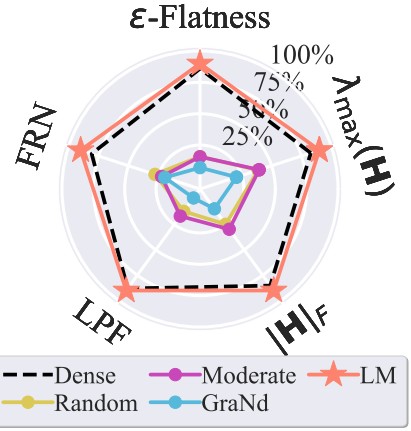

Figure A4: Flatness evaluations on the models pretrained with the pruned source data using different methods. The models are first pretrained and then finetuned using LP. The flatness are evaluated with respect to the downstream training loss and are quantified by the reversed sharpness evaluated through five widely acknowledged sharpness metrics. The results are normalized to $0\% \sim 100\%$, with $100\%$ denoting the highest (best) flatness across all the methods given one specific flatness metric.

the ViT structure. In line with our SSL experiment plan detailed in Tab. 2 of the paper, **Tab. A3** tested FM on three downstream datasets, specifically OxfordPets, Flowers102, and SUN397, with a source data class pruning ratio ranging from $50\%$ to $80\%$. The results affirm that the principal conclusions drawn from MoCov2 remain consistent with MoCov3 on ViT. Our method, FM, successfully identifies data subsets within the source dataset that can be pruned at high ratios without compromising downstream performance (termed as "winning subsets"). For instance, in one particular case, the FM-based winning subsets achieve a pruning ratio of up to $70\%$ on OxfordPets.

**Explore LM in the multi-task setting.** A limitation of our method is its task-specific nature in data influence analysis. As our methods assess source data influence for specific downstream tasks, developing a universal DP solution for multiple tasks simultaneously is challenging. As a preliminary study, we examine the performance of source dataset pruning for multiple downstream tasks simultaneously in **Fig. A5**. While LM can still identify winning subsets, the maximum pruning ratio diminishes as more tasks are considered. We will include this limitation in the Conclusion section.

**LM helps remove data biases through DP for a downstream task.** To investigate the scenarios with data biases, we conducted experiments on CIFAR-10C (the out-of-distribution scenario) [118]. We first pruned ImageNet given CIFAR-10 as a downstream task and evaluated the model on CIFAR-10C with different corruption types. **Tab. A4** shows LM results with different pruning ratios for 5 strong perturbation types in CIFAR-10C. Impressively, LM can achieve winning subsets with pruning up to $80\%$, even better than on CIFAR-10, confirming our method's effectiveness to filter biased data points to some degree.

Table A3: **Experiments with MoCov3 on ViT**. Other settings follow Tab. 2 of the original submission. The performance surpassing the unpruned setting (pruning ratio $0\%$) is highlighted in cyan . The best result in each setting is marked in **bold**. FM consistently outperforms other baselines and can find winning subsets with pruning ratios of more than 50%.

| Dataset | | OxfordPets | | | |
| --- | --- | --- | --- | --- | --- |
| Pruning Ratio | 0% | 50% | 60% | 70% | 80% |
| RANDOM | | 82.13 | 80.27 | 75.42 | 68.34 |
| MODERATE | 87.34 | 86.17 | 85.29 | 84.01 | 81.32 |
| GRAND | | 87.21 | 86.11 | 83.19 | 80.78 |
| FM (ours) | | **87.68** | **87.51** | **87.39** | **84.14** |
| Dataset | | SUN397 | | | |
| Pruning Ratio | 0% | 50% | 60% | 70% | 80% |
| RANDOM | | 59.22 | 58.45 | 56.73 | 54.29 |
| MODERATE | 60.36 | 60.13 | 59.61 | 58.21 | 56.44 |
| GRAND | | 60.39 | 59.27 | 58.95 | 57.15 |
| FM (ours) | | **60.49** | **60.55** | **60.42** | **59.88** |
| Dataset | | Flowers102 | | | |
| Pruning Ratio | 0% | 50% | 60% | 70% | 80% |
| RANDOM | | 92.41 | 91.65 | 90.17 | 88.41 |
| MODERATE | 93.96 | 93.75 | 92.41 | 91.42 | 90.11 |
| GRAND | | 93.88 | 93.21 | 91.77 | 90.45 |
| FM (ours) | | **94.11** | **94.28** | **93.97** | **91.42** |

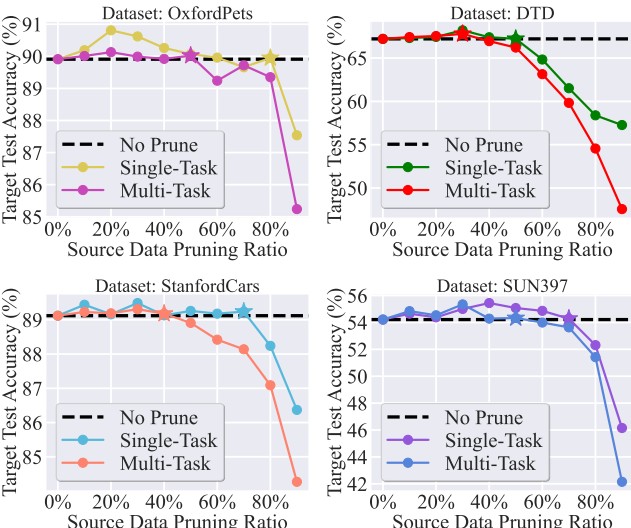

Figure A5: **DP achieved by LM in the multi-task setting given 4 downstream tasks**. This expands Fig. 4, *i.e.*, the single-task setting, where source data is pruned based on an individual task.

**Experiments on the few-shot benchmark VTAB [119].** We extend our experiments to include the VTAB benchmark [119]. As shown in Tab. A5, even in the few-shot setting, our major conclusions remain valid. LM can effectively pinpoint the most influential source subsets and eliminate the less valuable ones. For most tasks considered, winning subsets with pruning ratios up to $40\%$ are successfully found.

**Subsets obtained by LM improve the flatness of loss landscape on downstream tasks.** Fig. A4 evaluates the flatness of the loss landscape of the model pretrained on an $80\%$-pruned source dataset and subsequently finetuned on the downstream OxfordPets dataset [56]. Better flatness in the loss landscape is typically associated with better transferability. We quantify this flatness using the measure of *reversed sharpness* calculated via five widely accepted metrics: $\epsilon$-sharpness [120], low pass filter-based measure (LPF) [121], the max eigenvalue of the Hessian ($\lambda_{\max}(\mathbf{H})$) [122], the Frobenius norm of the Hessian ($|\mathbf{H}|_F$) [122], and Fisher Rao Norm (FRN) [123]. The results suggest that pretraining on the subset selected by LM leads the model towards a flatter region in

Table A4: Experiments on `CIFAR-10C`. LM-based source dataset pruning on ImageNet (given `CIFAR-10` as the downstream task) applies to transfer learning against `CIFAR-10C`. 5 out of the 19 corruption types are tested.

| Dataset | Pruning Ratio | | | | |
|---|---|---|---|---|---|
| | 0% | 20% | 40% | 60% | 80% |
| CIFAR10 | 96.83 | 96.88 | **97.03** | 96.57 | 95.41 |
| + Gaussian Noise | 82.13 | 82.67 | **82.89** | 82.60 | 81.19 |
| + Defocus Blur | 84.73 | 85.22 | **85.36** | 84.92 | 82.75 |
| + Impulse Noise | 84.62 | 85.21 | 84.78 | **85.93** | 85.11 |
| + Shot Noise | 83.18 | 83.25 | 83.49 | **83.76** | 83.24 |
| + Speckle Noise | 83.11 | **83.59** | 83.29 | 83.57 | 82.27 |

Table A5: Experiments on the few-shot transfer learning benchmark VTAB. Seven tasks in the `NATURAL` set are studied following the setting of Fig. 4. Each task contains 800 training and 200 testing samples.

| | Caltech101 | CIFAR-100 | DTD | Flowers102 | OxfordPets | SUN397 | SVHN | Mean |
|---|---|---|---|---|---|---|---|---|
| No Prune | 80.23 | 46.39 | 62.48 | 90.39 | 88.42 | 33.32 | 87.24 | 69.78 |
| Pruning 20% | **80.39** | **46.65** | 62.88 | 90.78 | 88.57 | **34.22** | **87.45** | **70.13** |
| Pruning 40% | 80.34 | 46.49 | **62.92** | **90.88** | **88.62** | 33.81 | 87.21 | 70.04 |
| Pruning 60% | 78.65 | 45.51 | 62.57 | 90.12 | 87.35 | 33.57 | 86.38 | 69.16 |
| Pruning 80% | 73.31 | 41.33 | 61.18 | 89.42 | 85.11 | 31.49 | 84.22 | 66.58 |

the downstream loss landscape, potentially contributing to the superior transferability of LM when compared to the baseline methods.

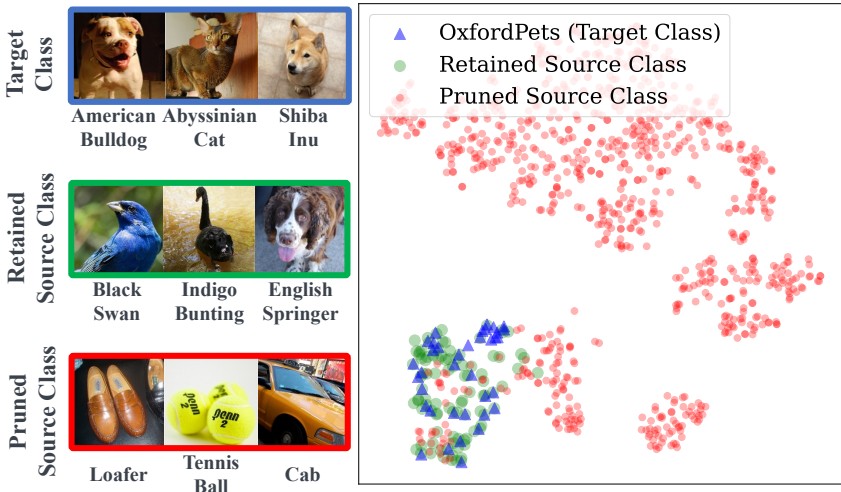

Figure A6: (Left) Interpretation merit of the data pruning strategy by LM. (Right) Feature distribution visualization using t-SNE for the source class selection by LM for OxfordPets with $90\%$ pruning ratio.

**Feature distribution analysis.** Fig. A6 provides visual explanations of DP at both deep representation (feature) and data levels. Here we focus on the LM method with a pruning ratio of $90\%$ given the downstream dataset OxfordPets. The other settings are consistent with Fig. 4. In **Fig. A6 (right)**, we visualize the source ImageNet classes (including pruned ones and retrained ones) and target OxfordPets classes in terms of their class-wise feature centroids in a 2D space achieved by t-SNE[124]. The class feature centroid is obtained by averaging the features of data points within a class, extracted by the pretrained source model on the full ImageNet dataset. As we can see, all retained source classes are grouped together and are close to the target classes. This indicates that the source classes that the pruning process share the most resemblance with the target data. In contrast, the pruned source classes are more dispersed and located further away from the target data classes. Furthermore, **Fig. A6 (left)** exhibits image examples of target classes as well as pruned and retrained source classes. We observe that image examples in the retained source classes (*e.g.*, relating to

Table A6: Performance on surrogate model size. Experiments follow the setting of Fig. 4: RN-101 is first pretrained on the pruned source dataset (ImageNet) based on the surrogate model, and then finetuned on the downstream task OxfordPets. Under different surrogate models, the source class selection overlapping ratio with the used surrogate model RN-18 in the submission is reported under $50\%$ pruning ratio.

| Surrogate Model Architecture | RN-20s | VGG-2 | RN-32s | VGG-4 | RN-44s | RN-56s | VGG-8 | RN-18 (Default) |
|---|---|---|---|---|---|---|---|---|
| Param. # (M) | 0.236 | 0.417 | 0.516 | 0.698 | 0.706 | 0.896 | 5.53 | 11.69 |
| Source Acc. (%) | 36.25 | 22.56 | 40.77 | 29.44 | 43.74 | 45.72 | 58.45 | 68.73 |
| Largest Pruning Ratio of Winning Subsets (%) | 60 | 50 | 80 | 70 | 80 | 80 | 80 | 80 |
| Source Class Selection Overlap (%) | 89.3 | 84.4 | 90.7 | 87.2 | 93.5 | 94.8 | 97.7 | 100 |

animals) are semantically closer to the target data points (relating to pets) than the pruned ones. This highlights the ability of LM to effectively identify and retain the most relevant classes for the downstream tasks. We provide more examples for FM in Fig. A7.

**Examining the top selected classes by FM and their image examples.** In **Fig. A7**, we showcase the top-10 source classes chosen by FM, as determined by the endowed scores. These selected classes closely correspond to the downstream datasets' subjects, demonstrating FM's effectiveness in identifying relevant classes for transfer learning. This finding also aligns with our observations in Fig. A6, showing that FM identifies source classes resembling downstream data.

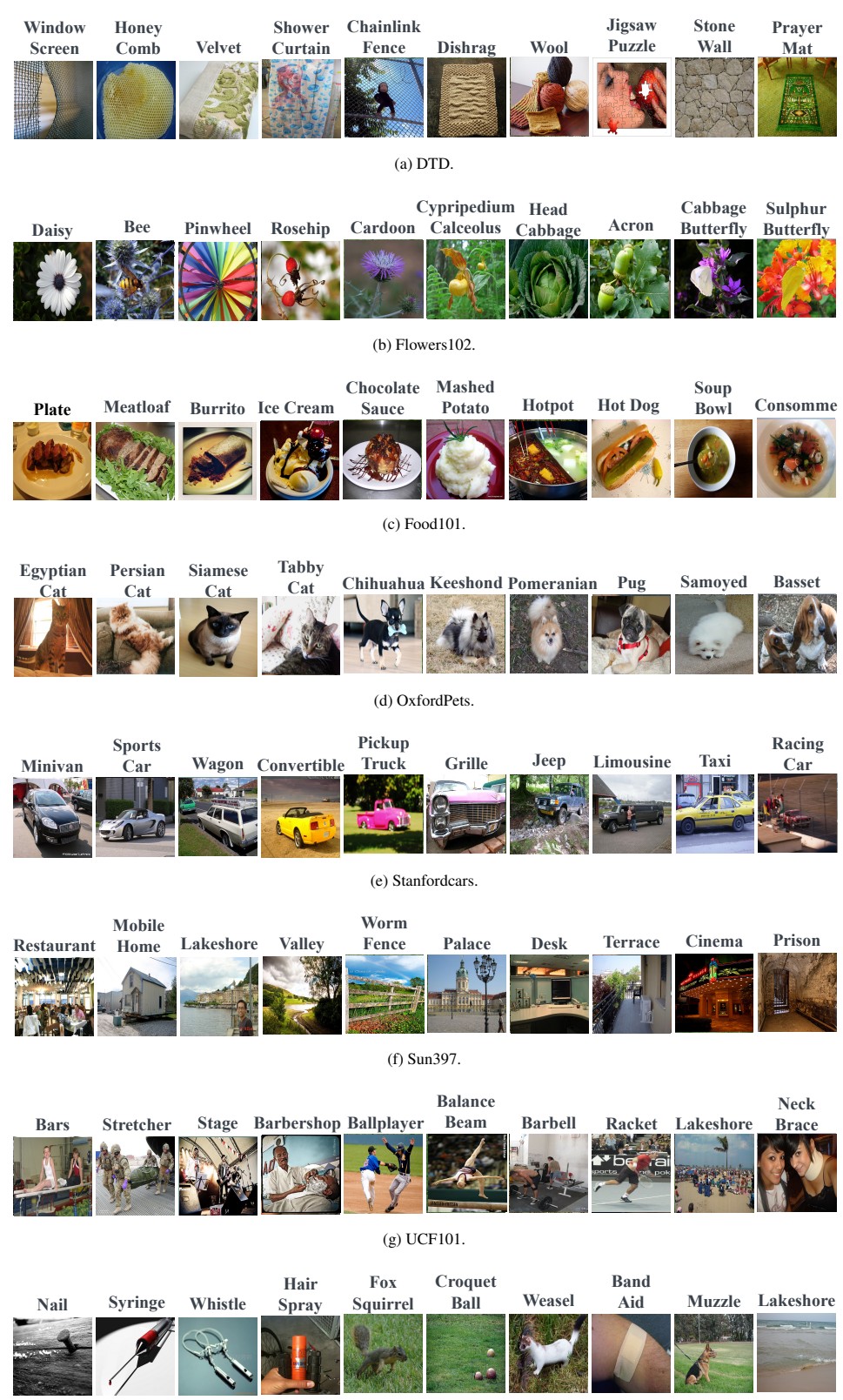

Figure A7: The source classes with top-10 scores selected by FM for the 8 downstream tasks studied in this work. For each class, the class label (name) as well as an representative image example is presented.

