# OpenReview forum: "Selectivity Drives Productivity: Efficient Dataset Pruning for Enhanced Transfer Learning"
_NeurIPS.cc/2023/Conference — NeurIPS 2023 poster_

### Official Review · Reviewer_vmFL · 2023-07-03

**Soundness:** 2 fair
**Presentation:** 2 fair
**Contribution:** 2 fair
**Rating:** 4
**Confidence:** 5

**Summary:**

The paper proposes two new dataset pruning (DP) methods, label mapping and feature mapping, for transfer learning. The aim is to improve pretraining efficiency and fine-tuning accuracy on downstream target tasks without sacrificing performance. The proposed methods are evaluated on numerous transfer learning tasks and show significant speed-up during the pretraining stage.

**Strengths:**

+ The paper addresses an important problem of dataset pruning for transfer learning, which has not been explored much in previous studies.

+ The proposed methods show significant improvement in pretraining efficiency and finetuning accuracy on downstream target tasks.

+ The paper provides a unified viewpoint to integrate DP with transfer learning.

+ The proposed methods are evaluated on numerous transfer learning tasks, demonstrating their broad applicability.

**Weaknesses:**

+ The paper does not consider other types of transfer learning scenarios, such as domain adaptation, multi-task learning, or meta-learning, which may have different challenges and requirements for dataset pruning.

+ The paper does not conduct a thorough ablation study or sensitivity analysis to understand the impact of different design choices, such as the surrogate model, the cluster number, and the reverse order selection, on the performance and efficiency of the proposed methods.

+ The paper does not provide any discussion of the limitations or failure cases of its method. This could limit the understanding of when and how to apply its method in practice.

+ The paper does not provide a detailed analysis of the impact of the proposed methods on the size of the pruned dataset, which could be an important factor in some applications.

+ The paper does not specifically address the question of whether it is necessary to compress data with a relatively small amount of data, but rather focuses on improving the efficiency of using the available data. It is important to note that while larger amounts of data can lead to better performance

+ The proposed DP methods require access to a surrogate source model (e.g., ResNet-18), which may not be available or practical in all settings.

+ The effectiveness of the proposed DP methods may depend on the quality and diversity of the source dataset, and it is unclear how well they would perform with low-quality or biased data.

+ The proposed DP methods (label mapping and feature mapping) were evaluated on a limited set of downstream tasks, and it is unclear how well they would generalize to other tasks or domains.

**Questions:**

Please see the weakness above, I would like the authors to address these limitations.

**Limitations:**

This article examines the use of transfer learning for data pruning, but questions whether this topic is meaningful. Firstly, the dataset used in the article is very small, with a maximum size of no more than 100,000, making it too small for practical application. Additionally, the current trend is to train very large models using large amounts of data. Furthermore, is the method used in the article simply a combination of two fields, namely data compression and transfer learning? It is important to research truly meaningful topics.

---

> ### Author Rebuttal · Authors · 2023-08-09
>
> Thank you for your constructive feedback. We provide response to **the eight points you listed in the weakness column (W1 to W8) and the limitations.** Additional results are indexed with "R" (e.g., Fig. R1) to distinguish them from the submission (e.g., Fig. 4), and are detailed in the one-page PDF.
>
> **Response to W1 & W8.**
>
> Thanks for the suggestions. We expanded our experiments to test our method's robustness, with details in **Fig. R1 and Tab. R2, R3, R4**. **Tab. R3** shows even with out-of-distribution biases, the LM method can prune over 40% of training data without affecting downstream performance, implying LM can enhance performance by removing harmful data. Tab. R4 highlights our method's success on the few-shot VTAB benchmark [R1], finding winning subsets with winning ratios up to 40%.
>
>  Besides, we’d also clarify that our paper's main focus is exploring data influence and DP for transfer learning. Our submission, covering two mainstream pretraining paradigms and eight datasets, have effectively demonstrated the efficacy and relevance of our proposed approach.
>
> &nbsp;
>
> **Response to W2.**
>
> Thanks. We'd like to clarify that our original submission **already explicitly includes sensitivity studies on cluster number and reverse order selection** (Lines 376-380, Fig. A2, A3 in Appendix).
>
> In addition, following your suggestion, we conducted sensitivity studies on surrogate model size (**Tab. R2 and Fig. R1**). Fig. R1 shows that the downstream performance of the original source pre-trained model RN-101 remains quite stable using diverse surrogate models (0.516M-11M in size), despite a reduction in the surrogate model's performance on the source dataset (Tab. R2). **Besides**, there is a strong agreement in source class choices across surrogate models of varying sizes. These results confirm our methods can adapt to models as small as about 1% the size of the original, emphasizing that superb performance with the surrogate model is not vital for obtaining source dataset pruning schemes with lossless downstream performance.
>
> &nbsp;
>
> **Response to W3.**
>
> Thanks for the insightful comments. A limitation of our method is its task-specific nature in data influence analysis. As our methods assess source data influence for specific downstream tasks, developing a universal DP solution for multiple tasks simultaneously is challenging.
> As a preliminary study, we examine the performance of source dataset pruning for multiple downstream tasks simultaneously (**Fig. R2**). While LM can still identify winning subsets, the maximum pruning ratio diminishes as more tasks are considered. We will include this limitation in the Conclusion section.
>
> &nbsp;
>
> **Response to W4 & W5**
>
> There might exist a misunderstanding of the motivation of our work. Please see general response ([GR1.1](./forum?id=D0MII7rP3R&noteId=JygqCsKyMH)) for a detailed clarification. (1) As shown in Fig. 4 and Tab. 2, DP can in fact enhance downstream performance, which is in line with the existing work [R2]. (2) DP could enable us to train the source model using an advanced but costly training protocol, such as adversarial training as we demonstrated in Line 347-367. (3) There also exist other strong reasons for the necessity of pruning the source dataset when public released models need to be debugged or updated to handle user-specific challenges, e.g., data regulation, security and privacy issues as suggested in [R2].
>
> &nbsp;
>
> **Response to W6**
>
> A5: Please see [GR1.3](./forum?id=D0MII7rP3R&noteId=JygqCsKyMH). As another interesting point, **Tab. R2 and Fig. R1** showed a remarkable level of agreement in the source dataset pruning among surrogate models of different sizes. See details in our **response to W2**.
>
> &nbsp;
>
> **Response to W7**
>
> A6: This question is insightful. To investigate the scenarios with data biases, we conducted experiments on CIFAR-10C (the out-of-distribution scenario) [R3]. We first pruned ImageNet given CIFAR-10 as a downstream task and evaluated the model on CIFAR-10C with different corruption types. Due to space limitations, Tab. R3 shows LM results with different pruning ratios for 5 strong perturbation types in CIFAR-10C. Impressively, LM can achieve winning subsets with pruning up to 80%, even better than on CIFAR-10, confirming our method's effectiveness to filter biased data points to some degree. Thank you for your valuable input!
>
> &nbsp;
>
> **Response to Limitation “This article examines the use of transfer learning for data pruning, but questions are whether this topic is meaningful. Is the method simply a combination of two fields, namely data compression and transfer learning?”**
>
> As shown in [GR1](./forum?id=D0MII7rP3R&noteId=JygqCsKyMH) (GR1.3), our study is practically meaningful and significant. We have made a notable contribution by tackling efficient source data pruning for transfer learning, a challenge previously unresolved. Our approach goes beyond a mere combination of dataset pruning and transfer learning. Line 176-194 and Fig. 2 demonstrate that a simple combination does not solve our studied problem.
>
> &nbsp;
>
> **Response to Limitation “The paper lacks analysis of the impact on the size of the pruned dataset. The dataset in the article is very small, with a max size of no more than 100,000, making it too small for practical application.”**
>
> We respectfully disagree that "the pruned dataset used is small, with a size of no more than 100,000." In our work, we adopt ImageNet as the source dataset to be pruned, with more than 1.3 million high-resolution images. ImageNet has been the most commonly used benchmark for pretraining and is the largest dataset considered in the literature [R2, R4].
>
> &nbsp;
>
> > [R1] A Large ... Visual Task Adaptation Benchmark
> >
> > [R2] A data-based perspective on transfer learning, CVPR 2023.
> >
> > [R3] Benchmarking neural network robustness to ..., ICLR 2019
> >
> > [R4] Coreset sampling from open-set for ..., CVPR 2023.

---

> > ### Comment · Reviewer_vmFL · 2023-08-11
> > **Dear Authors,**
> >
> > Thank you for your response, and I acknowledge that I have read it.
> >
> > For your response to W1 & W8. I think that the experiments conducted so far are still insufficient. Although the author has conducted numerous experiments, there is still a lack of diversity in the tasks. In my opinion, the author needs to continue conducting experiments, as I mentioned earlier. **You should consider other types of transfer learning scenarios, such as domain adaptation, multi-task learning, or meta-learning, which may have different challenges and requirements for dataset pruning.**
> >
> > So I think you should address this concern

---

> > > ### Author Response · Authors · 2023-08-11
> > > **Follow-up on Additional Experiment Datasets and Settings**
> > >
> > > Dear Reviewer vmFL,
> > >
> > > Thank you for your feedback. We are fully open and prepared to conduct further experiments as you suggested. However, it would greatly assist us if you could provide more specific guidance on the datasets and experimental setups that would best evaluate the effectiveness of our method.
> > >
> > > We feel that "domain adaptation" and "meta-learning" may not align well with the scope of our study on transfer learning. For example, domain adaptation typically involves training across different environments without a strict "source pretraining" phase, as demonstrated in DomainBed [R1].
> > >
> > > To clarify, **were you suggesting that we assess the performance of source dataset pruning (on ImageNet) within the context of transfer learning, possibly by treating domain adaptation as a downstream task? Additionally, if you could elaborate on whether you meant considering a meta-dataset as the target for transfer learning, we would greatly appreciate it as well.**
> > >
> > > Moreover, in our earlier response, **we presented a study on multi-task learning in Figure R2. If this doesn't align with your comment on "multi-task learning," we kindly ask for more detailed clarification.**
> > >
> > > We look forward to your guidance and greatly value your continued evaluation of our work.
> > >
> > > Thanks
> > >
> > > > [R1] In Search of Lost Domain Generalization.

---

> > > ### Author Response · Authors · 2023-08-14
> > > **Additional Experiment Results on Domain Generalization and Meta-Learning**
> > >
> > > Dear Reviewer vmFL,
> > >
> > > In response to your recent comments, we have taken the initiative (based on our best understanding) to conduct additional experiments to investigate the scenarios of domain generalization and meta-learning. Specifically, our focus has been on evaluating the efficacy of our proposed method, LM, using the DomainBed [R1] and Omniglot [R2] benchmarks.
> > >
> > >
> > > ### Domain Generalization
> > >
> > > We considered the VLCS [R3], PACS [R4], DomainNet [R5], and OfficeHome [R6] datasets in the additional experiments. Here one domain (e.g., domain "Cartoon" in PACS) was set aside for out-of-distribution (OOD) evaluation, and the model was trained on the remaining domains. OOD domain settings are revealed in **Tab. R7**. The transfer learning setting follows Fig. 4 in our submission: It involves pruning the source dataset (ImageNet) using our proposed method. This pruned dataset is then used to enhance the training of the source model (RN-101), followed by finetuning on the target downstream dataset. We consider both in-domain and cross-domain (OOD) evaluations of the finetuned model.
> > >
> > > Below are the tables showcasing our results. Notably,   our proposal exhibits remarkable capability in identifying valuable segments of the source dataset, efficiently eliminating unnecessary elements, even in the context of domain generalization tasks.
> > > Specifically, our method succeeded in removing about 40% of the source  dataset, leading to improvements in both the in-domain (ID) and OOD performance in the downstream tasks.
> > >
> > > **Tab. R6**: In-domain evaluation vs. source dataset pruning ratios using LM. Performance exceeding the unpruned results are marked in **bold**.
> > > | Dataset\Pruning Ratio |  0%  |    20%   |    40%   |    60%   |  80% |
> > > |:---------------------:|:----:|:--------:|:--------:|:--------:|:----:|
> > > |          VLCS         | 85.4 | **86.8** | **85.9** |   82.1   | 76.5 |
> > > |          PACS         | 95.4 | **95.9** |   95.2   |   94.1   | 92.8 |
> > > |       DomainNet       | 69.1 | **70.8** | **69.7** |   68.2   | 65.3 |
> > > |       OfficeHome      | 79.4 | **80.5** | **80.1** | **79.5** | 76.2 |
> > > |        Average        | 82.3 | **83.5** | **82.7** |   81.0   | 77.7 |
> > >
> > >
> > > **Tab. R7**: Cross-domain (OOD) evaluation  vs.  source dataset pruning ratios of LM. Performance exceeding the unpruned results are marked in **bold**.
> > > | Dataset\Pruning Ratio |  0%  |    20%   |    40%   |  60% |  80% | OOD Domain |
> > > |:---------------------:|:----:|:--------:|:--------:|:----:|:----:|:----:|
> > > |          VLCS         | 62.9 | **64.3** | **63.1** | 60.8 | 52.3 | Caltech101 |
> > > |          PACS         | 78.6 | **79.3** | **78.8** | 75.5 | 72.3 | Cartoon |
> > > |       DomainNet       | 61.3 | **62.9** | **61.8** | 60.1 | 55.2 | Painting |
> > > |       OfficeHome      | 69.8 | **71.2** | **70.5** | 68.4 | 65.5 | Art |
> > > |        Average        | 68.1 | **69.4** | **68.6** | 66.2 | 61.3 | - |
> > >
> > > ### Meta-Learning
> > >
> > > We implemented our method for the meta-learning task on Omniglot dataset and iMAML [R7] approach. Initially, the ImageNet dataset was pruned at different ratios, ranging from 10% to 50%, to enhance the training of the source model. This model was then used as the starting point for the subsequent meta-learning process, with all other settings following the iMAML framework.
> > >
> > >
> > > Remarkably, our experiments revealed that a pruning ratio of 30% enabled effective pruning without any loss in performance, highlighting the efficacy of our approach even in the context of meta-learning.
> > >
> > >
> > >
> > > **Table R8**: Meta-learning evaluation results on Omniglot using iMAML, comparing different source dataset pruning ratios of LM. "N-way-M-shot" refers to a classification task where N different classes are classified using M samples per class. Performance exceeding the unpruned results are marked in bold.
> > > | Pruning Ratio | 20-way-1-shot | 20-way-5-shot |
> > > |:----:|:---:|:---:|
> > > |0% (baseline) | 97.24% |98.92%|
> > > |      10%      |   **97.85%**  |   **99.32%**  |
> > > |      20%      |   **97.79%**  |   **99.25%**  |
> > > |      30%      |   **97.52%**  |   **99.18%**  |
> > > |      40%      |     96.84%    |   **98.98%**  |
> > > |      50%      |     96.23%    |     98.42%    |
> > >
> > >
> > > ### Conclusion
> > > In order to address the reviewer's concerns regarding additional downstream tasks, we conducted an additional set of experiments in the domain generalization and meta-learning contexts. We hope that these new results can enhance the confidence in the adaptability and effectiveness of our proposal.
> > >
> > > Thank you for your comments and consideration.
> > >
> > > > [R1] In Search of Lost Domain Generalization
> > > >
> > > > [R2] Human-level concept learning through probabilistic program induction.
> > > >
> > > > [R3] Unbiased metric learning: On the utilization of multiple
> > > datasets and web images for softening bias.
> > > >
> > > > [R4] Deeper, broader and artier domain generalization.
> > > >
> > > > [R5] Moment matching for multi-source domain adaptation.
> > > >
> > > > [R6] Deep hashing network for unsupervised domain adaptation.
> > > >
> > > > [R7] Meta-Learning with Implicit Gradients.

---

> ### Author Response · Authors · 2023-08-19
> **Two Days Left before Author-Reviewer Discussion Ends**
>
> Dear Reviewer vmFL,
>
> Thank you once again for your dedicated review of our paper. The insightful suggestions and thoughtful questions you've provided have been immensely valuable to us.
>
> As the discussion phase draws to a close with only 2 days remaining, we wish to extend a respectful request for your continued feedback about our most recent responses. Your perspective is of immense importance to us, and we eagerly anticipate any additional insights you might offer. If you find that these recent additions have played a role in mitigating some of your concerns, we would be deeply honored by your acknowledgment, and we hope to see a positive impact on the evaluation score.
>
> Furthermore, should any queries arise or if you require further clarifications, please do not hesitate to reach out. We are fully dedicated to providing timely and comprehensive responses throughout this pivotal phase of discussion.
>
> We deeply appreciate the time and effort you've invested in the review of our work.
>
> Best regards,
>
> Authors

---

> ### Comment · Reviewer_vmFL · 2023-08-21
> **Response**
>
> Thank you very much for the authors' response and additional experiments. Their efforts have addressed some of my concerns, and I appreciate that. I have decided to keep my score. However, I still feel that this paper lacks thoroughness in terms of experiments and theory. For example, there is still a lack of comprehensive experiments in areas such as domain adaptation, multi-task learning, or meta-learning, although the authors have added some experiments. Additionally, from a theoretical perspective, the paper does not provide a generalization analysis of the experimental methods. At this point, I still cannot fully support accepting this paper. Nevertheless, I have decided to keep my score and hope that the authors will continue to improve this paper further.

---

> > ### Author Response · Authors · 2023-08-21
> > **Further Response and A Confusion from Authors**
> >
> > Dear Reviewer vmFL,
> >
> > We appreciate your acknowledgment of our attempts to address your concerns, and thanks for recognizing our efforts in conducting additional experiments. Your constructive comments have been invaluable to us. In light of this, we would like to offer further responses in the hope that these will help alleviate the remaining concerns.
> >
> > **Response to "lacks thoroughness in terms of experiments"**
> >
> > We respectfully disagree with this assessment. As previously highlighted, our work focuses on efficiently evaluating source data influence for transfer learning and pruning less useful source classes to enhance transfer learning, as duly noted by reviewers RSvb, BXTc, and omyc. In line with this objective, we have conducted a comprehensive set of experiments both in our original submission and during the rebuttal phase, as also acknowledged by reviewers RSvb, BXTc, and omyc. Therefore, our paper is NOT weak in experimentation.
> >
> > **Response to "lacks thoroughness in terms of theory"**
> >
> > We greatly appreciate your insightful comment. However, we maintain a strong stance that conducting a theoretical generalization analysis of dataset pruning for transfer learning is beyond the scope of our current work. It's important to highlight that even within the domain of non-transfer learning, undertaking a generalization analysis of dataset pruning is inherently complex. We have acknowledged this as a prospective avenue for future research, which is underscored in the limitations section of our original submission (Line 388-389). On a related note, in the appendix, we have endeavored to provide additional insights into the generalization implications of dataset pruning through the lens of loss landscape flatness, utilizing five distinct flatness metrics (Figure A4, Line 756-765). We hope this supplementary perspective could help alleviate your concerns regarding the theoretical aspect.
> >
> > **Authors' confusion about the mismatched comments**
> >
> > We extend our gratitude for your continued engagement and valuable insights. However, we find ourselves perplexed by the inconsistency within the reviewer's recent comments. Specifically, on **August 20 at 6:56 PM PT**, the reviewer expressed, "Thank you very much for the authors' response and additional experiments. Their efforts have addressed some of my concerns, and I appreciate that. **I have decided to raise my score**. However, ..." Yet, within just a few hours (**August 21 at 12:50 AM PT**), these comments were modified to "Thank you very much for the authors' response and additional experiments. Their efforts have addressed some of my concerns, and I appreciate that. **I have decided to keep my score**. However, ..."
> >
> > The aforementioned alteration in decision has led to uncertainty regarding the underlying reasons. If there are any specific inquiries or concerns, we remain dedicated to addressing them diligently.
> >
> > Warm regards,
> >
> > Authors

---

### Official Review · Reviewer_omyc · 2023-07-04

**Soundness:** 3 good
**Presentation:** 2 fair
**Contribution:** 2 fair
**Rating:** 5
**Confidence:** 3

**Summary:**

This paper addresses the problem of dataset pruning, which serves to smaller the dataset for faster computation and learning. The authors claim that some existing methods require overly much computation for selecting the pruned dataset and thus propose a method with less computational requirements. However, in line 163 the authors seem to suggest pruning dataclasses from the source dataset, which requires an additional training of the foundation model, which is usually not possible in practice.

**Strengths:**

* The proposed method does not require training or heavy computation when selecting the source classes. Moreover, the results in Figure 4 suggest that better test accuracy is achieved for higher pruning rates in a wide range of datasets.

* Line 282-292 provides a clear overview of the training setup, which makes the experiments reproducible.


**Weaknesses:**

* The results in Figure 4 indicate that over a wide range of datasets about 70% of classes can be pruned. However, in Table 3 one can read that such pruning only saves about 3 hours of computation time. In what scenario is such saving critical? I would say that dataset pruning is complex and introduces many unknown biases, and I am not sure that this is outweighed by saving only three hours of computation.

* Removing any datapoint from the dataset introduces, by definition, a bias in the dataset. What are the insights on such biases and has any analysis been done on class imbalances? Alternatively, has any analysis on datasets such as CIFAR-C and ImageNet-C been run to assess robustness [1,2]?

* I would say the pruning of classes is specific to the target dataset. And usually, training of foundation models is not possible as the foundation models are usually trained by large institutions on large datasets and thus only fine tuning is an option. This paper seems to address the problem of pre-training models on a new dataset. What benefits does such an approach have over established foundation models? There have been papers to suggest that actually the pre-training on large datasets yields benefits for downstream performance [3].

[1] Hendrycks, Dan, and Thomas Dietterich. "Benchmarking neural network robustness to common corruptions and perturbations." arXiv preprint arXiv:1903.12261 (2019).

[2] https://github.com/hendrycks/robustness/blob/master/ImageNet-C/create_c/make_cifar_c.py

[3] Kolesnikov, Alexander, et al. "Big transfer (bit): General visual representation learning." ECCV, 2020.

**Questions:**

Have any experiments been done on benchmarks for transfer learning, such as [1] or [2]?

[1] Zhai, Xiaohua, et al. "The visual task adaptation benchmark." (2019).

[2] Triantafillou, Eleni, et al. "Meta-dataset: A dataset of datasets for learning to learn from few examples." arXiv (2019).


**Limitations:**

The paper focuses on computational efficiency and thus could reduce power consumption of learning algorithms at scale.

---

> ### Author Rebuttal · Authors · 2023-08-09
>
> We thank the reviewer for the constructive comments. **We use [W] and [Q] to refer to the specific points listed in the “Weakness” and “Question” columns in your official review respectively.** All the additional results use the index starting with “R” (e.g., Tab. R2, Fig. R1) to differentiate from the submission (e.g., Fig. 4) and are listed in the one-page pdf.
>
> **Q1: [W] Even though in Fig. 4 about 70% of the source classes can be pruned, the time saving (reduction) of 3 hours in Tab. 3 does not seem to be significant. In what scenario is such saving critical?**
>
> A1: Thank you for drawing attention to this aspect. We are afraid that our previous presentation might have led to a potential misunderstanding. Our experiments on supervised training (Fig. 2, 3, 4) followed the literature [R1], where we utilized the fast-forward accelerating framework (FFCV [R2]) to accelerate training on ImageNet (sorry for making this detail unclear). As shown in Tab. 3, FFCV allows us to train a ResNet101 on ImageNet within 6 hours. Aided by FFCV, the absolute time saving thus does not seem significant. However, it is crucial to consider the relative improvement. In this context, achieving a 56% computation time saving for a pruning ratio of 60% without losing downstream performance is indeed a significant accomplishment from our perspective.
>
> Further, we remark that FFCV is not a generic accelerated framework applicable to diverse settings. For example, it is not applicable in the self-supervised training (SSL) setting. In such a case, the absolute improvement in time saving provided by our proposal also becomes significant. In **Tab. R5** of the attached PDF, we showcase the time comparison related to Tab. 2 (SSL experiments), revealing that our method can save up to 168 hours (7 days) per single experiment trial by pruning 60% data without hurting the downstream performance for all the datasets in Tab. 2.
>
> Thus, the time savings from our method are indeed significant, especially when FFCV is not an option. Your observation has helped us recognize the need for a more detailed demonstration, and we will ensure this is addressed in our revision. Thank you!
>
> &nbsp;
>
> **Q2: [W] Dataset pruning is complex and introduces many unknown biases. Does the removal of data classes introduce additional biases? How should we analyze this? Are there any analyses on datasets such as CIFAR-C and ImageNet-C?**
>
> A2:  This is an insightful question. As detailed in [General Response 1 (GR1)](./forum?id=D0MII7rP3R&noteId=JygqCsKyMH), our work focuses on selecting the most influential source data points without sacrificing downstream performance. As [R1] shows, source data pruning can effectively remove the harmful influence of biased data classes relevant to the targeted downstream task, rather than introducing an additional data bias. This phenomenon is also confirmed in our work, where LM/FM successfully find the most useful points (Fig. 4), resulting in higher downstream performance than using the full data.
>
> To further justify this point, we followed the reviewer’s suggestion and investigated the applicability of our method on CIFAR-10C [R3]. The experiment setting is consistent with Fig. 4 based on the CIFAR-10 task and evaluated it on the CIFAR-10C with different corruptions. Due to space limit, in **Tab. R3** we report the results of LM on the 5 out of the 19 perturbation types using the largest perturbation strength defined in CIFAR-10C. As we can see, when facing distribution shifts, LM can obtain winning subsets with a pruning ratio up to 80%, even higher than DP for the CIFAR-10 task.
>
> In conclusion, our work may help removing data biases through dataset pruning for a downstream task, rather than inadvertently introducing them. The additional analyses on CIFAR-C reinforce our method's validity in handling these complexities. Your insightful query has greatly assisted us in this explanation, and we will include these discussions in our revision. Thank you!
>
> &nbsp;
>
> **Q3: [W] This paper seems to address the problem of pre-training models on a new dataset. What benefits does training a foundation model have over using the established foundation models?**
>
> A3: Thanks for this question. Our objective is NOT to address the problem of pretraining models on a new dataset. Instead, our primary objective is to study the influence of source data in transfer learning and to explore if an efficient dataset pruning algorithm can be designed for achieving lossless transfer learning on a given downstream task, as opposed to the computationally-intensive source data attribution method in [R1]. We kindly refer to [GR1.1](./forum?id=D0MII7rP3R&noteId=JygqCsKyMH) for the doubts about our setup, and [GR1.3](./forum?id=D0MII7rP3R&noteId=JygqCsKyMH) for practical significance and benefits of our proposal.
>
> &nbsp;
>
> **Q4: [Q] Have any experiments been done on benchmarks for transfer learning, such as [R4] and [R5]?**
>
> A4: Thank you for this insightful question! We have extended our experiments to include the VTAB benchmark [R4]. As shown in **Tab. R4**, even in the few-shot setting, our major conclusions remain valid. LM can effectively pinpoint the most influential source subsets and eliminate the less valuable ones. For most tasks considered, winning subsets with pruning ratios up to 40% are successfully found. Due to the time and space constraints, we did not report the results on [R5] and we will continue to run these additional experiments upon receiving the reviewer's follow-up response.
>
> &nbsp;
>
> > [R1] A data-based perspective on transfer learning, Jain et al., CVPR 2023.
> >
> > [R2] FFCV: Accelerating Training by Removing Data Bottlenecks, Leclerc et al., CVPR 2023
> >
> > [R3] Benchmarking neural network robustness to ..., ICLR 2019
> >
> > [R4] A Large ... Visual Task Adaptation Benchmark
> >
> > [R5] Meta-dataset: A dataset of datasets for learning to learn from few examples.

---

> > ### Comment · Reviewer_omyc · 2023-08-16
> > **Reply to rebuttal**
> >
> > Thank you for the rebuttal. A few more questions:
> >
> > From the rebuttal I read that the ‘work focuses on selecting the most influential source data points’, while from line 163 of the paper I understood that the method concerns pruning classes. They seem two different concepts. Does this method concern pruning classes or individual data points?
> >
> > About the compute savings, I understand that for the setup mentioned in the rebuttal, about half of the compute time on source training can be saved. How does that specific experiment compare on a data set like CIFAR-C? My main worry would be about the trade-off between compute saving and the biases or decreases in robustness introduced.
> >
> > Table R4 in the rebuttal is instructive to see that transfer learning on this wide range of datasets improves or has comparable results when pruning with the proposed method.

---

> > > ### Author Response · Authors · 2023-08-16
> > > **Thank you and further response**
> > >
> > > Dear Reviewer omyc:
> > >
> > > We thank you for your continued engagement and for raising further questions. We value your feedback and would like to provide clarifications to address your concerns:
> > >
> > > **Pruning Classes vs. Individual Data Points**
> > >
> > > We sincerely apologize if there was any confusion in our explanation. Our work focuses on pruning the source classes (for the supervise learning setting) and source data clusters (which are not necessarily the precise classes for self-supervised learning). We referred to both of these scenarios as "data points" in our communication. Our setup is consistent with the previous work [R1], where the focus was on source classes as the units of data influence analysis within the supervised learning context. As an illustration, we demonstrated that pruning the source classes could enhance transfer learning performance for downstream tasks, which aligns with the findings in Figure 2 of [R1]. However, it's important to note that the process of identifying and removing these bottom influential source classes is computationally demanding in [R1], as highlighted in the motivation section of our paper (Lines 144-154). We greatly appreciate your meticulous examination and will certainly improve the clarity of our explanations in the revised manuscript. Thank you for bringing this to our attention!
> > >
> > > **Compute Savings and Robustness on CIFAR-C**
> > >
> > > To elucidate on CIFAR-10C, we would like to clarify our experiment settings first. To the best of our knowledge, CIFAR-10C is a test set used for evaluation only. Therefore, the experiments associated with CIFAR-10C (**Tab. R3**) did not involve additional training. We utilized models from Fig. 4 of the CIFAR-10 dataset for evaluation purposes. Specifically, the ImageNet classes are first pruned (with CIFAR-10 being the downstream task), and the model (trained on the pruned ImageNet) is then finetuned on CIFAR-10. The resulting finetuned model is further evaluated on CIFAR-10C. Thus, the time savings calculation for CIFAR-10C mirrors that of CIFAR-10, as presented in **Tab. 3**. Regarding your concerns on the trade-off between time savings and the biases introduced, as shown in Tab. R3, compared to the original evaluation of transfer learning on CIFAR-10C (i.e., 0% pruning ratio in Tab. R3), our method leads to improved robustness with a pruning ratio of up to 60%~80%. This is not at the cost of taking higher computation time.
> > >
> > > **Table R4 shows the proposed method is effective on a wide range of datasets.**
> > >
> > > We are glad you found value in our additional experiments. We'll incorporate these results and associated discussions in the revision. Your constructive feedback is invaluable to us.
> > >
> > > Should there be any additional queries or concerns, please do not hesitate to communicate. We are committed to ensuring clarity and precision in our work.
> > >
> > > Thank you once again for your time and insights. We hope our first round and the latest responses have addressed your concerns appropriately.
> > >
> > > > [R1] A data-based perspective on transfer learning, Jain et al., CVPR 2023.

---

> ### Author Response · Authors · 2023-08-20
> **One Day Left before Author-Reviewer Discussion Ends**
>
> Dear Reviewer omyc,
>
> We thank you once again for your diligent review and valuable feedback. In our last response to your latest comments, we provided additional clarifications on our dataset pruning setting and the computation savings with CIFAR-10C. As the discussion phase draws to a close with only 1 day remaining, we would greatly appreciate your consideration in possibly raising the original rating (5) if you find our response satisfactory. However, if you believe that there are any remaining areas where additional clarifications/responses could contribute to such a higher rating, please do not hesitate to inform us. Your guidance is always crucial to improve the quality of our submission!
>
> Warm regards,
>
> Authors

---

> > ### Comment · Reviewer_omyc · 2023-08-22
> >
> > Thanks for the clarification on the rebuttal. Table R3 shows that even under pruning, as described in the paper, the robustness on CIFAR-C stays on par with non-pruning approaches. This addresses a concern I raised in the review. I encourage the authors to include such results in the paper.

---

### Official Review · Reviewer_BXTc · 2023-07-06

**Soundness:** 2 fair
**Presentation:** 3 good
**Contribution:** 2 fair
**Rating:** 6
**Confidence:** 3

**Summary:**

This paper presents the problem of dataset pruning for transfer learning. The idea is to remove redundant samples in the source dataset to improve the source pre-training efficiency while maintaining the downstream fine-tuning accuracy. The label and feature mapping method proposed for the problem utilizes a surrogate model to compute a relatedness score for each source class. Source classes with lower scores are then pruned afterward. The proposed methods are compared to existing dataset pruning methods and demonstrate better performance on 8 downstream datasets.

**Strengths:**

- The paper is well-written and easy to follow. The proposed problem is interesting and the proposed approaches seem to be reasonable.
- The experimental study on adversarial pre-training is inspiring and shows the potential for a wider extension of the problem to other areas.

**Weaknesses:**

- The problem assumes that the source pre-training dataset is available during downstream fine-tuning. However, this might not be realistic because the pre-training dataset is sometimes private and inaccessible, like medical domains.
- The efficiency comparison with pre-training on the unpruned dataset is a little bit unfair to me (Table 3). In the standard pre-training and fine-tuning paradigm, we only need to pre-train once, and then we can fine-tune the pre-trained model on multiple tasks. However, in the proposed scenario, we need to pre-train on the pruned dataset for each downstream task every time. Pre-training on the unpruned dataset yields a generic model but requires more time. Although pre-training on the pruned dataset is faster, it outputs a task-specific model. It would be important to distinguish the trade-off to make a fair comparison.
- The need for training a surrogate model might be a huge limitation because it cannot avoid training on the whole source dataset, which could be extremely large. Although the authors try to argue that the surrogate model can be small, it remains unclear how small it can be.

**Questions:**

Please see the above weakness section and also the following questions.

1) By assuming the availability of the pre-training dataset during fine-tuning, what would be the practical use cases for the proposed problem? What could be the problem’s additional cost and benefits, compared to the existing pre-training and fine-tuning paradigm, where we just need the pre-trained model, without the pre-training dataset?

2) Table 3 only compares the time consumption for one downstream task. However, the scenario might become different if we consider multiple downstream tasks at once because we only need to pre-train once for the unpruned dataset. Given some downstream tasks, like 8 tasks in the paper, what would be the downstream accuracy if we use the same time budget for pre-training on pruned or unpruned datasets?

3) How accurate does the surrogate model need to be for the proposed methods? For example, if we use a surrogate model even smaller than ResNet-18 and get a lower pre-training accuracy, how would the fine-tuning accuracy change?

**Limitations:**

As mentioned in the conclusion of the paper, the paper mainly focuses on the experimental results of the proposed methods and lacks theoretical viewpoints/insights. Besides, the use of a surrogate model and the assumption of an accessible pre-training dataset might be some other potential limitations.

---

> ### Author Rebuttal · Authors · 2023-08-09
>
> We thank the reviewer for the detailed feedback. We present a point-to-point response below, where **we use [W] and [Q] to refer to the specific points listed in the “Weakness” and “Question” columns in your official review.** All the additional results use the index starting with “R” (e.g., Tab. R2, Fig. R1) to differentiate from the submission (e.g., Fig. 4) and are listed in the attached pdf.
>
> **Q1: [W] [Q] The assumption of accessible pretraining dataset might sometimes be unavailable, because the source dataset could be private. [Q] What would be the practical use cases for the proposed problem and what are the benefits of the proposed method over using the off-the-shelf pretrained model?**
>
> A1: Thanks for raising this valuable question. We apologize for any confusion introduced by our previous presentation. However, it appears there may be a misunderstanding regarding our method's motivation and setting. Please refer to the [general response (GR1.1)](./forum?id=D0MII7rP3R&noteId=JygqCsKyMH) for detailed clarifications.
>
> As a highlight, since our primary motivation is to study the source data influence in transfer learning, we need access to the pre-training dataset to be pruned and knowledge of the downstream task. Although this differs from some transfer learning scenarios where only the pretrained model is available, it provides a fair setting for dataset pruning, consistent with [R1].
>
> Moreover, instead of using a once-for-all pre-trained source model, there exist practical use cases that the off-the-shelf pre-trained source model needs to be customized for user-specific challenges, e.g., distribution shifts (see **Tab. R3**) or other data regulation, security and privacy issues as suggested in [R1], or the need for more advanced but computationally-intensive training protocol, such as adversarial training as we showed in the experiment section (see Line 347 ~ Line 367).
>
> Last but not the least, our approach is not positioned in opposition to the trend of using off-the-shelf source models. Rather, it seeks to bridge the gap between the capabilities of these open-source models and the computationally-intensive source model modifications for particular tasks, industries, or organizations. We strongly believe that our method makes a novel contribution to data-model influence in transfer learning.
>
> > [R1] A data-based perspective on transfer learning, Jain et al., CVPR 2023.
>
> &nbsp;
>
> **Q2: [W] [Q] Unfair efficiency comparison in Tab. 3. Model pretrained on the full source dataset can be used for all downstream tasks, while model pretrained on pruned source dataset can only be applied to one task only. How about comparing the training time for several downstream tasks?**
>
> A2: This is an important point! We acknowledge that a model pretrained on a pruned source dataset might not be as versatile as one pretrained on the full dataset. However, the context of our approach caters to exploring the source data influence in a specific downstream task, which we have elaborated on in [General Response 1 (GR1.3)](./forum?id=D0MII7rP3R&noteId=JygqCsKyMH).
>
> Your comment prompted us to further investigate the applicability of our method across multiple downstream tasks. In **Fig. R2**, we demonstrate LM's performance when the source dataset is pruned collectively for four downstream datasets. Interestingly, LM maintains its efficacy in this expanded setting, successfully discovering source subsets without compromising downstream performance (termed "winning subsets") with a maximum pruning ratio ranging from 30% ~ 50%. Compared to the results in Fig. 4, the largest pruning ratio is reduced as more downstream tasks are considered in LM. This also suggests that the task-specific customization of the source model allows for a larger margin for dataset pruning.
>
> We appreciate this insight, recognizing it as a limitation, and will address and discuss it appropriately in our revised manuscript.
>
> &nbsp;
>
> **Q3: [W] [Q] It is not clear how small the surrogate model can be and how accurate the surrogate model needs to be? The usage of the surrogate model could be a limitation of this work.**
>
> A3: Thank you for this constructive and insightful question. To explore the sensitivity of our methods to the surrogate model's size and accuracy, we have conducted additional experiments using various surrogate models. These results are presented in **Tab. R2** and **Fig. R1**. In accordance with Fig. 4 in our paper, we replaced ResNet-18 with smaller neural networks, specifically ResNet20s, ResNet32s, ResNet44s, ResNet56s, VGG-2, VGG-4, and VGG-8. **Tab. R2** shows the transfer learning performance on the task OxfordPets against different surrogate model sizes. It is evident that even though the performance of the surrogate model on the source dataset (ImageNet) decreases, the downstream performance of RN-101 pretrained on the LM-based pruned source subsets remains relatively stable.
>
> To further elucidate this observation, **Tab. R2** compares the class indices selected by the corresponding surrogate models. Interestingly, the most relevant source class selections exhibit a high level of agreement across surrogate models of differing sizes. Our label mapping method  shows robust behavior with different surrogate models, even if they yield different performance on the source dataset due to their different model capacities. **For example, ResNet32s only achieves a test accuracy of 40.77% on the source dataset, but it manages to achieve the same maximum pruning ratio of the winning subsets as ResNet-18 on the downstream task OxfordPets.** This experiment underscores that our method can accommodate a small surrogate model of 1% the size of the pretrained model (Rn-32s vs. RN-101). Moreover, it reveals that the surrogate model does not need to attain exceptional performance in the source domain for dataset pruning in transfer learning.

---

> ### Author Response · Authors · 2023-08-19
> **Two Days Left before Author-Reviewer Discussion Ends**
>
> Dear Reviewer BXTc,
>
> We extend our heartfelt appreciation for your dedicated review of our paper. As outlined in the recent NeurIPS email communication, we are fully aware of the commitment and time your review entails. Your efforts are deeply valued by us.
>
> With only 2 days remaining before the conclusion of the discussion phase, we wish to extend a respectful request for your  feedback about our general response (including 1-page PDF) and individual responses. Your insights are of immense importance to us, and we eagerly anticipate your updated evaluation. Should you find our responses informative and useful, we would be grateful for your acknowledgment. Furthermore, if you have any further inquiries or require additional clarifications, please don't hesitate to reach out. We are fully committed to providing additional responses during this crucial discussion phase.
>
> We sincerely thank you for your continued support and consideration. Your expertise is pivotal to the advancement of our research.
>
> Best regards,
>
> Authors

---

> > ### Comment · Reviewer_BXTc · 2023-08-20
> >
> > Thanks for the detailed response. I have also read all the other reviews and author responses.
> >
> > I can better understand the significance of the problem now. I suspect my previous confusion might come from the introduction section (Line 21-26), which makes me feel the paper is opposing the generic large-scale pre-training. I would suggest the authors enhance it by incorporating the General Response in the future version.
> >
> > Thanks to the authors for providing the additional experiments. It's nice to see the multi-task results in Figure R2. That said, I still feel it's a bit misleading to directly compare the time of generic pre-training (with unpruned dataset) and customized pre-training (with pruned dataset) in Table 3 and in the corresponding paragraph (Line 332-346). Even in the multi-task setting of Figure R2, the model pre-trained on the pruned dataset is still specific to the 4 downstream tasks. However, with my current understanding, I wonder if the arguments in Line 332-346 are necessary. In my opinion, if the paper is mainly for customized pre-training, there seems to be no need to claim that it is more efficient than generic pre-training. Not sure if I understand correctly. Can the authors explain a bit this part of the paper?
> >
> > Finally, the experimental results in Figure R1 and Table R2 have addressed my concern about the accuracy of the surrogate models.

---

> > > ### Author Response · Authors · 2023-08-20
> > > **Thank You and Further Response**
> > >
> > > Dear Reviewer BXTc,
> > >
> > > We would like to extend our sincere thanks to your time and patience for reading our responses and other reviews. We are also grateful for your latest insightful comments.
> > >
> > > **Introduction Clarification**
> > >
> > > We are encouraged to see that our response was uselful to alleviate your concerns on the significance of our studied problem. Following your suggestion, we will for sure revise the introduction part (especially Lines 21-26) and incorporate our general response to make our statement clearer. In particular, we will emphasize that this work is not set opposed to the generic foundation model training. Thank you!
> > >
> > > **Runtime Efficiency Comparison Concern: Table 3 & Line 332-346**
> > >
> > > Firstly, your understanding about the efficiency of the customized pretrain built upon dataset pruning is correct. We also understand the reviewer’s concern and thus, **we have decided to revise Table 3 and the discussion in Line 332-346**. Before we elaborate on our revision plan, we would also like to clarify our original intention to disclose the computation time comparison. We intended to use Table 3 to demonstrate that “our dataset pruning method and the involved surrogate model training will not introduce significant computational overheads”. In light of your feedback, **we plan to update Table 3 (and the relevant discussions in Line 332-346) to the following Table R5**, where we will shift our comparison focus to **the computation costs of dataset pruning methods**, showcasing the efficiency of our approach. In particular, we compare our approach with the important prior work [R1] and two SOTA dataset pruning baselines (as shown in Figure 2) Moderate Coreset [R2] and GraNd [R3]. As we can see from Table R5, our method (even using a surrogate model) achieves a substantial computation efficiency improvement over other methods.
> > >
> > > **Table R5**. Time consumption comparison of different dataset pruning methods. The time consumption is evaluated at the pruning ratio of 60%. Other settings are the same as Table 3 in the original submission.
> > > |  Pruning Method  | [R1] | [R2] | [R3] | Ours |
> > > |:----------------:|:----:|:----:|:----:|:---------:|
> > > | Time Consumption (h) | >500 |  7.6 | 18.4 |    2.4    |
> > >
> > > In particular, we would like to mention that [R1] adopts a brute-force method and involves training thousands of source models. Thus, its computational cost is typically unaffortable in experiments.
> > >
> > > **Surrogate Models Feedback**
> > >
> > > We are glad to see that our additional results alleviate your concerns on the surrogate models. We will for sure include such discussions in the revised manuscript.
> > >
> > > In summary, we hope our response has drawn a clear picture regarding the efficiency discussion. If you find our responses satisfactory, we would also greatly appreciate your consideration in possibly raising the original rating. However, if you believe there are any remaining areas where additional clarifications/responses are needed, please don't hesitate to inform us. Your guidance is always crucial to improve the quality of our submission.
> > >
> > >
> > > > [R1] A data-based perspective on transfer learning, CVPR 2023.
> > > >
> > > > [R2] Moderate Coreset: A Universal Method of Data Selection for Real-world Data-efficient Deep Learning, ICLR 2023
> > > >
> > > > [R3] Deep Learning on a Data Diet: Finding Important Examples Early in Training, NeurIPS 2021.

---

> > > > ### Comment · Reviewer_BXTc · 2023-08-20
> > > >
> > > > Thank the authors for their response.
> > > >
> > > > I can understand the authors' original intention for Table 3 now. Maybe the word "enhance" in Line 332 and Line 340 is a little bit too direct, making me think that the authors want to claim "DP is more efficient than generic pre-training" in a direct comparison. Shifting the comparison focus to the computation costs of related DP methods is better and makes more sense to me. Thank you for providing these additional results.
> > > >
> > > > I have no further questions from my side. Since my concerns have been addressed, I would like to raise the score to 6.

---

> > > > > ### Author Response · Authors · 2023-08-20
> > > > > **A Thank You Note to Reviewer BXTc**
> > > > >
> > > > > Dear Reviewer BXTc,
> > > > >
> > > > > Thank you so much for your prompt response! We highly appreciate your feedback and your decision to increase the score to 6. Your acknowledgment is encouraging and holds significant value for us. We will make sure to revise our paper in accordance with your comments and suggestions!
> > > > >
> > > > > Best regards,
> > > > >
> > > > > Authors

---

> > > > > ### Author Response · Authors · 2023-08-21
> > > > > **Thank You and A Kind Remind of Review Score Update**
> > > > >
> > > > > Dear Reviewer BXTc,
> > > > >
> > > > > We sincerely appreciate your consideration to adjust the rating to a 6! However, we noticed that this change has not yet been reflected in the official review score. If possible, we would appreciate a lot if you could update it before the discussion phase concludes. Your continued support means a lot to us. Thank you again for your time and support!
> > > > >
> > > > > Warm regards,
> > > > >
> > > > > Authors

---

### Official Review · Reviewer_RSvb · 2023-07-11

**Soundness:** 4 excellent
**Presentation:** 3 good
**Contribution:** 3 good
**Rating:** 7
**Confidence:** 3

**Summary:**

The following paper tackles the problem of dataset pruning from the lens of transfer learning, where the main goal is to achieve lossless or improved transfer learning accuracy of the source model on a target task using less amount of source data. To achieve this, this paper proposes two dataset pruning algorithms, Label Mapping (LM) for supervised pretraining and Feature Mapping (FM) which is capable of finding winning subsets within source dataset for self-supervised pre-training by monitoring the model's responsiveness toward target data samples. Experiments show that incorporating LP and FM on the source dataset by 40-80% pruning ratio does not sacrifice the classification performance on downstream tasks while speeding up the pretraining process by a factor of 2-5. Furthermore, additional experiments on adversarial pretraining tasks show classification performance improvement of 1-2% on downstream tasks.

**Strengths:**

(+) The Paper is well-written and easy to understand.

(+) The methods are highly intuitive, simple, and efficient, making them accessible for general-purpose usage.

(+) Experiments show promising results on various benchmarks with regards to various metric of interest not limited to downstream task performance, and pre-training time, and leads to a more efficient adversarial learning

**Weaknesses:**

(-) For self-supervised learning experiments, I think the experiment is quite limited, where only one self-supervised method and one network architecture is used for evaluation. Perhaps adding more relevant self-supervised learning architectures such as MoCov3 [1], which uses ViT, or information-maximization self-supervised learning methods such as Barlow Twins [2] would strengthen the paper.

(-) Minor typos:

a) In pg. 2, finetue -> fine-tune

b) In pg. 7, savings -> saved

**Questions:**

As pointed in Weakness, would the transfer learning results based on self-supervised learning pretraining holds for Feature Mapping (FM) on other self-supervised learning methods, i.e. MoCov3 [1] or Barlow Twins [2]?

[1] https://arxiv.org/abs/2104.02057

[2] https://arxiv.org/abs/2103.03230

**Limitations:**

Yes.

---

> ### Author Rebuttal · Authors · 2023-08-09
>
> We sincerely appreciate your constructive comments and a precise understanding of our contributions. **We use [W] and [Q] to refer to the specific points listed in the “Weakness” and “Question” columns in your official review correspondingly.** Please see our response below.
>
> **Q1: [Q] [W] More experiments are suggested on more recent self-supervised learning architectures, such as MoCov3 with ViT or Barlow Twins to strengthen the paper.**
>
> A1: Thank you for your valuable suggestions. In response, we have conducted new experiments to illustrate the effectiveness of our proposed method (FM) when applied to the more recent SSL framework MoCov3 using the ViT structure. Due to the limited time available during the rebuttal phase, we were only able to conduct experiments with MoCov3. However, we recognize the importance of extending our work to include other recent frameworks like Barlow Twins, and we will continue to run these additional experiments upon receiving the reviewer's follow-up response. Detailed results of the experiments conducted so far can be found in **Tab. R1** in **the attached PDF** of the general response.
>
> In line with our SSL experiment plan detailed in Tab. 2 of the paper, **Tab. R1** tested FM on three downstream datasets, specifically OxfordPets, Flowers102, and SUN397, with a source data class pruning ratio ranging from 50% to 80%. The results affirm that the principal conclusions drawn from MoCov2 remain consistent with MoCov3 on ViT. Our method, FM, successfully identifies data subsets within the source dataset that can be pruned at high ratios without compromising downstream performance (termed as "winning subsets"). For instance, in one particular case, the FM-based winning subsets achieve a pruning ratio of up to 70% on OxfordPets.
>
> &nbsp;
>
> **Q2: [W] There are some minor typos.**
>
> A2: We appreciate your meticulous attention to detail. Rest assured, we have corrected these minor typos in our revision. Thank you for bringing them to our attention.

---

> > ### Comment · Reviewer_RSvb · 2023-08-19
> > **To Authors**
> >
> > Thank you for providing us with additional experiment results on downstream tasks. With this, I have acknowledged the contributions made by the authors and would like to keep my score.

---

> > > ### Author Response · Authors · 2023-08-19
> > > **Thank You for Your Acknowledgement on Our Contribution!**
> > >
> > > Dear Reviewer RSvb,
> > >
> > > We would like to extend our gratitude again for your valuable feedbacks as well as for acknowledging the contributions of this work. This is encouraging! We will certainly integrate your suggestions in the revision of our paper.
> > >
> > > Best,
> > >
> > > Authors

---

### Author Rebuttal · Authors · 2023-08-09

## Highlighted General Response

We thank all the reviewers for their valuable comments. First, we would like to clarify some possible misunderstandings about our work based on the comments.

### GR1 - Possible misunderstandings of the motivation and the setting of this work. (Reviewer BXTC, vmFL, omyc)


A few concerns were raised by reviewers on the availability of the source dataset (**@Reviewer BXTc**) or the surrogate model (**@Reviewer vmFL**), as well as the necessity of pretraining own models instead of using off-the-shelf pretrained foundation models (**@Reviewer omyc, vmFL**). It seems that the motivation and the primary objective of our work was not very clearly delivered. We apologize for that! **This work aims to study the influence of source data in transfer learning and to explore an efficient data pruning algorithm designed for lossless transfer learning given a downstream task**, as opposed to the computationally-intensive source data attribution method for transfer learning in [R1]. We highlight our motivation and setting below.

* **GR1.1: A data-based perspective of transfer learning, but in a much more efficient manner (@Reviewer BXTc, omyc, vmFL).** Our work follows the existing research in data influence analysis for transfer learning [R1, R2], where the most influential source data is pinpointed given a downstream task. As our paper discussed (Line 54 - 58), [R1] pioneered this line of research and proposed a brute-force approach to evaluate the influence of every source class. Such a manner is not affordable when the problem scales up. This motivates a more efficient source data influence analysis method, which is scalable to diverse downstream tasks and to different pre-training paradigms (as we showed in supervised learning, adversarial pre-training, and SSL scenarios). **However, this is also a non-trivial problem as the existing dataset pruning (DP) methods become ineffective in transfer learning (see Fig. 2 vs. Fig. 4).** As our primary motivation is to study source data influence in transfer learning, we naturally require access to the pre-training dataset to be pruned and knowledge of the downstream task. This provides a fair setting for DP, consistent with [R1].

* **GR1.2: The use of a smaller source surrogate model is not a limitation but an interesting and important aspect from the perspective of DP (@Reviewer vmFL).** Previous DP studies assumed no mismatch between DP and model training on the pruned dataset. However, our work demonstrates for the first time that DP can be conducted with a much smaller surrogate model without sacrificing downstream task performance. **This brings a significant learning efficiency benefit, as selecting the most influential source data can now be achieved in an ultra-efficient manner, unlike the super-costly approach in [R1].** Moreover, we find that the assumption of having access to a surrogate small model is not restrictive. The results show that **our approach is NOT sensitive to the surrogate model size**, accommodating a small surrogate model of nearly 1% the size of the original pretrained model without downstream performance loss (see **Tab. R2** and **Fig. R1**).

* **GR1.3: The study of source dataset pruning for transfer learning has practical significance and is a meaningful and prosperous field. (@Reviewer vmFL, BXTc, omyc).** In response to why we should conduct source dataset pruning and retrain our own model instead of using public ones, there are compelling reasons. For example, public released models may not meet the needs for user-specific challenges, e.g., distribution shifts (see **Tab. R3**) or other data regulation, security and privacy issues as suggested in [R1], or need to be improved using another more advanced but computationally-intensive training protocol, such as adversarial training as we demonstrated in the experiment section (Line 347-367). Thus, our proposal can speed up the customized source training process and reduce the computational costs. Furthermore, investigating source data influence to predict and manage the behavior of models, including data-models [R3], is a promising field that enhances understanding of data biases and optimizes resource allocation. This work resonates with existing studies [R1, R2], continuing an established line of inquiry acknowledged for its value in the community .

In summary, our work is not opposed to the current transfer learning paradigm, but a focused and efficient solution tailored to specific needs with broader implications, offering valuable insights into data influence and source training customization.

> [R1] A data-based perspective on transfer learning, CVPR 2023.
>
> [R2] Coreset sampling from open-set for fine-grained self-supervised learning, CVPR 2023.
>
> [R3] Datamodels: Predicting Predictions from Training Data, ICML 2022

---

### GR2 - A summary of additional experiments in the one-page PDF.

We have made a substantial effort to enrich our experiments (see the attached PDF). Below is a summary of them, where Q$i$ represents the $i$th question in our response).

* Reviewer RSvb
    * Q1 - Experiments on advanced SSL frameworks, MoCov3, with ViT (**Tab. R1**).


* Reviewer BXTc
    * Q2 - Experiments in the multi-task setting (**Fig. R2**)
    * Q3 - Sensitivity study of the surrogate model size (**Fig. R1**, **Tab. R2**).


* Reviewer omyc
    * Q1 - Time saving report for SSL (**Tab. R5**).
    * Q2 - Experiments on CIFAR-10C (**Tab. R3**).
    * Q4 - Experiments on transfer learning benchmark VTAB (**Tab. R4**).


* Reviewer vmFL
    * Q1, Q6 - Experiments on CIFAR-10C (**Tab. R3**).
    * Q1 - Experiments in the multi-task setting (**Fig. R2**).
    * Q1 - Experiments on transfer learning benchmark VTAB (**Tab. R4**).
    * Q2, Q5 - Sensitivity study of the surrogate model size (**Fig. R1**, **Tab. R2**).

---

### Decision · Program_Chairs · 2023-09-21

**Decision:**

Accept (poster)

**Comment:**

I am recommending to **accept** this paper "Selectivity Drives Productivity: Efficient Dataset Pruning for Enhanced Transfer Learning".

Four reviewers provided feedback on this paper. The authors provided a response to the reviews and I appreciate the authors' comments and clarifications, specifically I think the authors were successful at discussing their work with the reviewers and at clarifying reviewers' concerns.

The reviewers did not fully agree on the assessmetn with scores of 4/5/6/7. Based on the reviews and the rebuttal discussions, the reviews with the slightly higher scores (5/6/7) carry somewhat more weight in my opinion.

This paper looks at an interesting combination of problems - dataset pruning and transfer learning. I share a concern about the practical relevance of the approach that has been voiced by reviewers, though.

Overall, I believe the paper is of high enough quality to be accepted at NeurIPS, hence my recommendation to accept.